# Clone-Robust AI Alignment

**Ariel D. Procaccia** [1]  **Benjamin Schiffer** [2]  **Shirley Zhang** [1]

## Abstract

A key challenge in training Large Language Models (LLMs) is properly aligning them with human preferences. Reinforcement Learning with Human Feedback (RLHF) uses pairwise comparisons from human annotators to train reward functions and has emerged as a popular alignment method. However, input datasets in RLHF can be unbalanced due to adversarial manipulation or inadvertent repetition. Therefore, we want RLHF algorithms to perform well even when the set of alternatives is not uniformly distributed. Drawing on insights from social choice theory, we introduce robustness to approximate clones, a desirable property of RLHF algorithms which requires that adding near-duplicate alternatives does not significantly change the learned reward function. We first demonstrate that the standard RLHF algorithm based on regularized maximum likelihood estimation (MLE) fails to satisfy this property. We then propose the weighted MLE, a new RLHF algorithm that modifies the standard regularized MLE by weighting alternatives based on their similarity to other alternatives. This new algorithm guarantees robustness to approximate clones while preserving desirable theoretical properties.

## 1. Introduction

As the reasoning capabilities of Large Language Models (LLMs) improve and as LLMs begin to play a larger role in society, it is increasingly important for LLMs to be aligned with human values. One common method used for AI alignment is Reinforcement Learning with Human Feedback (RLHF). In RLHF, a human annotator is typically shown two answers to a prompt, and asked to report which answer they prefer. This process is repeated across many annotators and potentially many types of questions and answers,

and results in a large dataset of pairwise comparisons. An RLHF algorithm then takes the pairwise comparison dataset as input and outputs a reward function that assigns values to answers. One reason why RLHF is an appealing technique is because of the ease of data elicitation, as it is simpler for humans to pick a favorite between two answers than it is to rank many answers or provide good 'role model' answers for the LLM.

Fundamentally, the mandate of RLHF algorithms is to solve a preference aggregation problem, where the goal is to find a reward function that best aligns with the values of the general population, given pairwise comparison data. This goal is complicated by the fact that humans often have diverse preferences, and may not agree on the best answer to a question. Luckily, the study of how to aggregate diverse preferences is not a new area in computer science, but one that has been long explored by researchers in social choice theory. Classic social choice considers settings with sets of voters and sets of alternatives, where each voter provides a ranking over alternatives. These rankings are then provided as input to a voting rule, which outputs a summary of the voters' preferences (such as a single winner or an overall ranking). Social choice studies the design and analysis of such voting rules.

In the RLHF setting, the 'voters' are the human annotators, who provide pairwise preferences over the 'alternatives'. There are several reasonable choices for how to define an 'alternative' in RLHF. Perhaps the simplest definition is that an alternative is just a single answer. However, many RLHF datasets in practice contain answers to multiple questions. Therefore, an alternative could also be a question-answer pair. Finally, answers (and questions) are often generated by various LLMs, so an alternative could also be viewed as the LLM model which generated the answer. Whichever the case, an RLHF algorithm would then be the 'voting rule' which takes as input the preference data and outputs a summary of the preferences – in this case, a single reward function. The close relationship between RLHF and voting theory means that we can take inspiration from past work in social choice to both anticipate potential pitfalls in RLHF and design better RLHF algorithms.

One especially relevant potential pitfall is that input datasets in RLHF are not necessarily balanced in the types of questions and answers that are included, whether due to random-

---

[1]Department of Computer Science, Harvard University [2]Department of Statistics, Harvard University. Correspondence to: Benjamin Schiffer <bschiffer1@g.harvard.edu>, Shirley Zhang <szhang2@g.harvard.edu>.

*Proceedings of the $42^{nd}$ International Conference on Machine Learning*, Vancouver, Canada. PMLR 267, 2025. Copyright 2025 by the author(s).

ness or adversarial influence. For example, at Anthropic, questions are generated by crowdworkers, who have the flexibility to communicate with LLMs on any topic of interest (Bai et al., 2022). In ChatGPT, answers are typically generated by LLMs, and depending on how the LLM was trained, some answers may look more similar than others (OpenAI, 2022). Because the generation processes for these datasets are not well-regulated, it would be ideal for an RLHF algorithm to not be sensitive to near duplicates and to perform well even on unbalanced datasets. This is important for at least two reasons. First, RLHF algorithms which are not robust to near duplicates will require more careful design of the input dataset, which may increase cost and restrict the types of questions and answers that can be included in the dataset. Furthermore, it may be more difficult to add to the dataset over time, as it will be necessary to check for near duplicates in the existing dataset. Second, RLHF algorithms which are robust to near duplicates will also be more robust to both adversarial manipulation and accidental duplication.

In social choice, if a voting rule is robust to adding duplicates of alternatives, it is said to satisfy *independence of clones*. Informally, a voting rule is independent of clones if after adding an alternative $a'$ which is equivalent to another alternative $a$, the output of the voting rule does not change. In the RLHF setting, we can think of *approximate clones* as two alternatives which are very close by a given distance metric and for which all annotators have very similar values, where the distance metric depends on the nature of the alternatives. For example, if the alternatives are textual responses, then a reasonable distance metric might be the Euclidean distance between their vector embeddings. An approximate clone of an textual response might look like the original response with an adjective replaced by a synonym. We can then extend the concept of independence of clones to *robustness to approximate clones*. Informally, an RLHF algorithm is robust to approximate clones if adding a new alternative to the data set that is similar to an existing alternative does not significantly change the output reward function. This is intuitively a desirable property because adding a new alternative that is very similar to an existing alternative does not provide much new information about annotator preferences.

Building on these insights, our main research goals are to

1. evaluate the robustness of current RLHF algorithms in the presence of approximate clones, and

2. develop RLHF algorithms which are robust to approximate clones.

### 1.1. Our Results

To address the first goal, we show that the standard RLHF algorithm, which uses the *regularized maximum likelihood estimator (MLE)*, is not robust to approximate clones. In

response to the second goal, we propose a new algorithm for RLHF that we call the *weighted MLE*. Intuitively, the weighted MLE adjusts the objective function of the MLE by down-weighting alternatives that are similar to other alternatives (and therefore provide less new information) and up-weighting alternatives that are different than other alternatives (and therefore provide more new information). Our main result is that the weighted MLE is robust to approximate clones; we also demonstrate that it retains many of the clean interpretations of the regularized MLE.

In addition to our main result about independence of clones, we also prove an impossibility result for RLHF in the presence of diverse preferences. We show that for any RLHF algorithm, there exists a population such that even with constant scaling, the distance between the RLHF algorithm output and the mean rewards of the population is arbitrarily large. We show this for a population that consists of a mixture of only two Bradley-Terry-Luce (BTL) models, thereby highlighting the inherent difficulty of aggregating preferences of populations with even simple diversity in preferences.

We also extend the ideas of Siththaranjan et al. (2023) that relate the output of the regularized MLE to the average win rates of the alternatives. We show that the output reward function of the regularized MLE is the solution to a system of equations, where the left hand side is the average win rates for the rewards output by the MLE and the right hand side is the empirical average win rates. We similarly show that the output of the weighted MLE has the same relationship to the empirical weighted average win rates.

We conclude with a case study using LLM generated answers to a single prompt. We use LLMs as annotators to generate two preference datasets, where one dataset has additional cloned alternatives. We then approximate both the standard MLE algorithm and the weighted MLE algorithm using neural networks. In this experiment, we show that the output of the standard MLE is significantly more affected by the presence of approximate clones than the output of the weighted MLE, which supports our theoretical results.

### 1.2. Related Work

RLHF has recently gained traction as a popular method of aligning LLMs with human preferences (Bai et al., 2022; Ouyang et al., 2022; Ziegler et al., 2019). The potential benefits of applying social choice theory to the RLHF setting have not gone unnoticed. Recent work has mapped classic social choice concepts to RLHF (Dai & Fleisig, 2024) and extended social choice axioms for RLHF (Ge et al., 2024), considered personalization as a way to address diversity (Poddar et al., 2024; Park et al., 2024), and studied other methods of aggregating diverse preferences (Zhong et al., 2024; Swamy et al., 2024).

Our work particularly focuses on the social choice concept of independence of clones, which was first introduced by Tideman (1987). Follow-up work has studied manipulation by cloning and clone structures (Elkind et al., 2010; 2012), and recent work has studied an even stronger notion, obvious independence of clones (Berker et al., 2025). In a position paper, Conitzer et al. (2024) propose various high-level directions for applying social choice to RLHF, and in particular identify independence of clones as a desirable property for RLHF algorithms, because chatbot responses may be very similar to each other. They point out that Borda count, a voting rule which is implicitly used in current approaches to RLHF (Siththaranjan et al., 2023), is not independent of clones. In our work, we elaborate on this insight by considering approximate clones and providing specific instances for which standard RLHF algorithms are not robust to approximate clones.

We highlight several papers that are especially related to ours, and include more details in Appendix A. Like us, Xu et al. (2023) are concerned about duplicates in answers shown to annotators, but unlike us, their results are primarily for dichotomy models and three-way comparisons. Also like us, both Siththaranjan et al. (2023) and Chakraborty et al. (2024) give different forms of impossibility results for RLHF algorithms in the presence of diverse preferences.

An independent work published shortly after this paper also studied the problem of clones and used a different weighting scheme to address the clone problem (Berriaud & Wattenhofer, 2025).

Further afield, recent papers have considered other forms of robustness in RLHF, such as robustness to incorrect or corrupted data (Bukharin et al., 2024; Mandal et al., 2024). In our work, we specifically focus on robustness to approximate clones, and expect inconsistent data due to diversity in the annotator population.

## 2. Model

Suppose we have a set of annotators $\mathcal{N} = [n]$ and an infinite set of all possible alternatives $\mathcal{S}$. We define $|\mathcal{S}|$ as the volume of $\mathcal{S}$ and assume that $|\mathcal{S}|$ is finite. Each alternative $s \in \mathcal{S}$ has a *context* $c(s) \in \mathbb{R}^d$ which represents important features of the alternative. For notational convenience, we will often refer to the context $c(s)$ simply by $s$. We only observe a finite subset of alternatives $\mathcal{M} = [m] \subset \mathcal{S}$. Each annotator has a reward function $r_i^* : \mathcal{M} \to \mathbb{R}$, where $r_i^*(x)$ represents the reward of annotator $i$ for alternative $x$.

Given two alternatives, an annotator expresses a preference over the two alternatives based on their reward function. As is common in RLHF, we assume that the expressed preferences of annotators follow a Bradley-Terry-Luce (BTL) model (Bradley & Terry, 1952; Bai et al., 2022), in that

an annotator $i$ states a preference for alternative $x_1$ over alternative $x_2$ with probability

$$p_i(x_1 \succ x_2) = \frac{e^{r_i^*(x_1)}}{e^{r_i^*(x_1)} + e^{r_i^*(x_2)}}.$$

The BTL model takes into account the fact that annotator preferences may be noisy or inconsistent across queries, especially when the reward gap of two alternatives is small. When annotators are drawn randomly, we then denote the expected probability of seeing $x_1$ preferred to $x_2$ as $p^*(x_1 \succ x_2) = \mathbb{E}_i[p_i(x_1 \succ x_2)]$.

We assume that annotator reward functions are Lipschitz continuous in the Euclidean distance between the context of two alternatives, as stated below.

**Assumption 2.1.** For all players $i \in \mathcal{N}$, the reward function $r_i^*$ is Lipschitz continuous with parameter $K > 0$. Formally, for any $i \in \mathcal{N}$ and any $x_1, x_2 \in \mathcal{S}$, $|r_i^*(x_1) - r_i^*(x_2)| \leq K \|x_1 - x_2\|_2$.

Let a query be a pairwise comparison $q = \{x_1, x_2\}$, where $x_1, x_2 \in \mathcal{M}$, and let a set of queries be denoted $\mathcal{Q}$. For every $x_1, x_2 \in \mathcal{M}$, we assume that $\{x_1, x_2\}$ is included in $\mathcal{Q}$ at least once. For $x_1, x_2 \in \mathcal{M}$ and $i \in \mathcal{N}$, define the random function $f_{\mathrm{BTL}}(\{x_1, x_2\}, i) \in \{x_1, x_2\}$ such that $\Pr(f_{\mathrm{BTL}}(\{x_1, x_2\}, i) = x_1) = p_i(x_1 \succ x_2)$. In other words, $f_{\mathrm{BTL}}(q, i)$ is one sample of annotator $i$'s preference between $x_1$ and $x_2$ according to annotator $i$'s true rewards for $x_1$ and $x_2$ in the BTL model. Further let $f_{\mathrm{BTL}}(q) = f_{\mathrm{BTL}}(q, i')$ when $i'$ is drawn uniformly at random from $\mathcal{N}$. A preference dataset $\mathcal{D}(\mathcal{Q})$ is generated from $\mathcal{Q}$ by choosing an annotator uniformly at random for each query and sampling that annotator's preference over the alternatives in that query, i.e. $\mathcal{D}(\mathcal{Q}) = \{(q, f_{\mathrm{BTL}}(q)) : q \in \mathcal{Q}\}$.

For a given preference dataset $\mathcal{D}$, define $p_{\mathcal{D}}(x_1 \succ x_2)$ as the proportion of time that $x_1$ is preferred to $x_2$ in $\mathcal{D}$. We say that a dataset $\mathcal{D}$ is *representative* if $p_{\mathcal{D}}(x_1 \succ x_2) = p^*(x_1 \succ x_2)$ for all $x_1, x_2 \in \mathcal{M}$. An RLHF algorithm ALG takes as input a preference dataset $\mathcal{D}$ and returns a reward function $r(\cdot)$ where $r : \mathcal{M} \to \mathbb{R}$. Note that ALG does not know any information about the annotators (such as $\mathcal{N}$, $p_i$, or $p^*$). The goal of ALG is to find a "good" reward function $r$ based on $\mathcal{D}$.

For intuition, it may be helpful to keep the following preference dataset generation example in mind:

**Example 2.2.** Suppose that we want to find a reward function $r$ which evaluates responses to a specific question $Z$. Then $\mathcal{S}$ would be the set of all possible responses to $Z$, and $\mathcal{M}$ would be a finite subset of responses to $Z$. We can generate each query $q \in \mathcal{Q}$ by randomly sampling two responses $x_1, x_2$ from $\mathcal{M}$. We can then generate a preference datum for this query by randomly sampling an annotator $i$ from $\mathcal{N}$ and asking annotator $i$ for their preference between $x_1$ and

$x_2$. The set of all preference datum then forms our dataset $\mathcal{D}$, which we give as input to an RLHF algorithm ALG.

## 2.1. MLE with Diverse Preferences

We first consider the setting with only one annotator ($n = 1$). When $n = 1$, the query data is generated from a single BTL model. A natural solution is to estimate the unknown reward function $r^*$ as the reward function that best matches the data in $\mathcal{D}$. Using the Kullback-Leibler divergence $\mathrm{KL}(\cdot)$ as the distance metric, the reward function that best approximates the data in $\mathcal{D}$ (minimizes the KL divergence) when $n = 1$ is

$$\hat{r}^1 := \arg\min_r - \sum_{x_1, x_2 \in \mathcal{M}} p_{\mathcal{D}}(x_1 \succ x_2) \cdot \log\left(\frac{e^{r(x_1)}}{e^{r(x_1)} + e^{r(x_2)}}\right).$$

When the number of samples for every pair of alternatives is the same, $\hat{r}^1$ is exactly the MLE solution for RLHF. In RLHF, maximum likelihood estimation refers to finding the rewards for the single BTL model that has the highest probability of generating the observed data.

Furthermore, because the true underlying distribution is a single BTL model when $n = 1$, standard MLE theory implies that $\hat{r}^1$ will converge to $r^* := \mathbb{E}_i[r_i^*] = r_1^*$ as the number of comparisons for each pair of alternatives goes to infinity (Zhu et al., 2023).

Having $\hat{r}^{\mathcal{D}}$ converge to $\mathbb{E}_i[r_i^*]$ is a natural goal in RLHF, as this means that the reward function converges to the mean rewards for the underlying population. Unfortunately, when $n > 1$, no RLHF algorithm can accurately recover $\mathbb{E}_i[r_i^*]$ for every possible population even if the algorithm is given infinite query data. This negative result holds even for $n = 2$ and $m = 2$ and is formally proven in Theorem 2.3 in terms of Euclidean distance. Note that there is an additive constant in this result because the BTL model is invariant to additive constants. Furthermore, Theorem 2.3 does not contradict the positive results of Zhang et al. (2022), as their results require sufficiently many alternatives in order to make the problem identifiable.

**Theorem 2.3.** *Let $n = 2$ and suppose $\mathcal{D}$ is a representative preference dataset over alternatives in $\mathcal{M}$. Then for any algorithm ALG and any $C > 0$, there exist $r_1^*$ and $r_2^*$ such that $r^{\mathcal{D}} := \mathrm{ALG}(\mathcal{D})$ satisfies*

$$\min_{\alpha \in \mathbb{R}} \sum_{x \in \mathcal{M}} \left(\frac{r_1^*(x) + r_2^*(x)}{2} - r^{\mathcal{D}}(x) - \alpha\right)^2 > C.$$

The proof of Theorem 2.3 can be found in Appendix B.

Even though accurately estimating the mean reward functions of the population is not always possible for $n > 1$, the same arguments that motivate using $\hat{r}^1$ in the $n = 1$ case can also be applied to the $n > 1$ case. More specifically, we can still find the single BTL model that best approximates $\mathcal{D}$. As is the case when $n = 1$, the single BTL model that minimizes the KL-divergence is also the MLE solution. When $n$ may be greater than 1, a regularization term must

also be included to guarantee that the optimization problem has a solution. Formally, for $\lambda > 0$, define

$$\hat{r}^{\mathcal{D}} := \arg\min_r - \sum_{x_1, x_2 \in \mathcal{M}} p_{\mathcal{D}}(x_1 \succ x_2) \cdot \log\left(\frac{e^{r(x_1)}}{e^{r(x_1)} + e^{r(x_2)}}\right)$$
$$+ \frac{\lambda}{2} \sum_{x \in \mathcal{M}} r(x)^2. \tag{1}$$

Using this regularized MLE is a standard method for RLHF due its interpretability (Siththaranjan et al., 2023). Note that a single BTL model is typically used to approximate $\mathcal{D}$ both for the simplicity of the model and because $n$ is unknown to the algorithm. Another benefit of the above formulation is that the objective in Equation (1) is strictly convex and has a unique global minimum (Siththaranjan et al., 2023). In practice, the optimization problem can be solved approximately using a sufficiently large function class for $r$ such as a neural network. In the following sections, we will analyze this standard MLE-based RLHF algorithm and propose a new algorithm with additional theoretical guarantees.

## 2.2. Average Win Rate and Borda Count

In addition to the interpretation discussed above, Siththaranjan et al. (2023) showed that the order in which the alternatives are ranked by the regularized MLE is the same as the order in which the alternatives are ranked by the average win rate. In Theorem 2.5, we show an even stronger relationship between the regularized MLE and the average win rate, which is that the regularized MLE is the unique solution to a system of equations involving the empirical average win rates. This gives additional interpretability to the regularized MLE, as the regularized MLE is therefore similar to an M-estimator where the $m$ moments correspond to the win rates of the $m$ alternatives.

**Definition 2.4.** For a dataset $\mathcal{D}$ over alternatives $\mathcal{M}$, the average win rate of alternative $x \in \mathcal{M}$ is

$$\mathrm{AWR}_{\mathcal{D}}(x) := \frac{1}{m} \sum_{y \in \mathcal{M}} p_{\mathcal{D}}(x \succ y).$$

**Theorem 2.5.** *Let $\hat{r}^{\mathcal{D}}$ be the regularized MLE as defined in Equation (1) and define*

$$\widehat{\mathrm{AWR}}(x) = \frac{1}{m} \sum_{y \in \mathcal{M}} \frac{e^{\hat{r}^{\mathcal{D}}(x)}}{e^{\hat{r}^{\mathcal{D}}(x)} + e^{\hat{r}^{\mathcal{D}}(y)}}.$$

*Then $\hat{r}^{\mathcal{D}}$ is the solution to the system of equations*

$$\mathrm{AWR}_{\mathcal{D}}(x) = \lambda \hat{r}^{\mathcal{D}}(x) + \widehat{\mathrm{AWR}}(x) \quad \forall x \in \mathcal{M}.$$

The proof of Theorem 2.5 can be found in Appendix F. Importantly, the average win rate as defined above is conceptually similar to the Borda count score in social choice

theory. Therefore, Theorem 2.5 implies a close relationship between the regularized MLE and the Borda count voting rule, a relationship first observed by Siththaranjan et al. (2023). The close relationship between the regularized MLE and Borda count is a key aspect of the proofs in the following sections.

## 3. Robustness to Approximate Clones

In this section, we adapt the concept of independence of clones from social choice to the RLHF setting. In traditional social choice, independence of clones is a desirable characteristic of voting rules which intuitively states that the winner of an election remains the same when duplicates of candidates are added to the candidate pool. The Borda count voting rule, which is closely related to the MLE in RLHF (see Section 2.2), does not satisfy independence of clones. See Appendix C.1 for a formal definition of independence of clones in traditional social choice.

We adapt the traditional independence of clones definition for RLHF. Informally, we say that an RLHF algorithm is robust to approximate clones if adding new alternatives that are clones of existing alternatives does not significantly change the reward function that is output by the RLHF algorithm. Note that robustness to approximate clones in RLHF guarantees stability of the reward function instead of merely the winner, and is therefore a stronger notion. As an RLHF algorithm only has access to noisy observations regarding human preferences, we will also only require reward function stability when we have representative datasets, or in other words, in cases when the empirical pairwise win rates are the same as the true pairwise win rates. If the dataset is not representative, it is not necessarily desirable that the reward function is unchanged when a clone is added because there may be value in generating a larger dataset. When the dataset contains sufficiently many queries, the empirical pairwise win rates will approximately equal the true pairwise win rates by the law of large numbers. Additional justification of our definition of robustness to approximate clones in RLHF can be found in Appendix C.2.

We are now ready to formally present our definition of robustness to approximate clones for RLHF.

**Definition 3.1** (Robust to Approximate Clones). An algorithm ALG is robust to approximate clones if for every $\mathcal{M} \subseteq \mathcal{S}$ and $\delta > 0$ there exists an $\epsilon > 0$ such that the following holds. Suppose $\mathcal{M}' = \mathcal{M} \cup \{x'\}$, where $x' \in \mathcal{S}$ and $\exists x \in \mathcal{M}$ such that $\|x - x'\| \leq \epsilon$. Let $\mathcal{D}$ be a representative dataset of queries over the alternatives $\mathcal{M}$ and let $\mathcal{D}'$ be a representative dataset of queries over the alternatives in $\mathcal{M}'$. Let $\hat{r} = \mathrm{ALG}(\mathcal{D})$ and let $\hat{r}' = \mathrm{ALG}(\mathcal{D}')$. Then $|\hat{r}'(x) - \hat{r}'(x')| \leq \delta$ and for all $x \in \mathcal{M}$, $|\hat{r}(x) - \hat{r}'(x)| \leq \delta$.

Informally, a RLHF algorithm satisfies Definition 3.1 if adding a new alternative whose context is "very close" to

that of an existing alternative does not significantly change the reward function. This is desirable because if the players' values are Lipschitz continuous, a new alternative whose context is very similar to that of an existing alternative provides little new information. Note that this can intuitively be viewed as requiring that the output reward function is continuous in the set of alternatives.

Robustness to approximate clones is also reminiscent of core ideas in differential privacy (Dwork, 2006), where the goal is to design algorithms that are robust to removing any data point. Any RLHF algorithm which satisfies robustness to approximate clones also automatically satisfies *exact independence of clones*, which informally says that adding new alternatives which are exact clones does not change the output reward function at all. We formally define and discuss exact independence of clones in Appendix C.3.

Importantly, the regularized MLE does not satisfy Definition 3.1, as stated formally in Theorem 3.2. See Appendix D for the proof of Theorem 3.2.

**Theorem 3.2.** *Let $\hat{r}^{\mathcal{D}}$ be the regularized MLE as defined in Equation (1). The algorithm $\mathrm{ALG}(\mathcal{D}) = \hat{r}^{\mathcal{D}}$ is not robust to approximate clones.*

## 4. Weighted MLE

In this section, we propose a modified version of the regularized MLE which satisfies Definition 3.1 while maintaining the interpretability inherent to the original MLE.

The main idea of the proposed algorithm is to modify the objective function by weighting each alternative by how unique that alternative is compared to the other alternatives in $\mathcal{M}$. Therefore, an alternative with context very similar to the context of other alternatives will have a smaller weight, while an alternative with a context very different than the context of other alternatives will have a larger weight. Informally, the weight of an alternative $y$ is the fraction of alternatives in $\mathcal{S}$ that are closer to $y$ than to any other alternative in $\mathcal{M}$ (with ties split evenly among all tied alternatives). We define the weights formally in Definition 4.1.

**Definition 4.1.** For any set of alternatives $\mathcal{M} \subseteq \mathcal{S}$ and any $x \in \mathcal{S}$, define $\mathrm{proj}_{\mathcal{M}}(x) = \arg\min_{y \in \mathcal{M}} \|x - y\|_2 \subseteq \mathcal{M}$. For $y \in \mathcal{M}$, define

$$w_{\mathcal{M}}(y) = \frac{1}{|\mathcal{S}|} \int_{x \in \mathcal{S}} \frac{1_{y \in \mathrm{proj}_{\mathcal{M}}(x)}}{|\mathrm{proj}_{\mathcal{M}}(x)|} dx.$$

Note that by this construction, $\sum_{y \in \mathcal{M}} w_{\mathcal{M}}(y) = 1$. Using these weights, we define the *weighted MLE* as $\hat{r}_{\mathrm{w}}^{\mathcal{D}} = \arg\min_r f_{\mathcal{D}}(r)$, where

$$f_{\mathcal{D}}(r) = -\sum_{x_1, x_2 \in \mathcal{M}} \left( w_{\mathcal{M}}(x_1) w_{\mathcal{M}}(x_2) p_{\mathcal{D}}(x_1 \succ x_2) \right.$$

$$\times \log \left( \frac{e^{r(x_1)}}{e^{r(x_1)} + e^{r(x_2)}} \right) \Bigg)$$

$$+ \frac{\lambda}{2} \sum_{x \in \mathcal{M}} w_{\mathcal{M}}(x) r(x)^2. \tag{2}$$

Intuitively, $f_{\mathcal{D}}(r)$ down-weights terms involving alternatives that provide less new information because they are very similar to other alternatives. Consequently, two alternatives that are approximate clones of each other will both have smaller weights.

We are now ready to state our main result, which is that the weighted MLE is robust to approximate clones

**Theorem 4.2.** *Under Assumption 2.1, the algorithm* $\mathrm{ALG}(\mathcal{D}) = \hat{\mathrm{r}}_{\mathrm{w}}^{\mathcal{D}}$ *is robust to approximate clones.*

While we defer the formal proof of Theorem 4.2 to Appendix G, we will next provide some intuition for why this result holds. Consider the Voronoi diagram of the alternatives in $\mathcal{M}$, which consists of a partitioning of $\mathcal{S}$ into regions, where each region corresponds to all of the points in $\mathcal{S}$ that are closest to some alternative in $\mathcal{M}$. For example, if $\mathcal{S} = [0, 1] \times [0, 1]$ and $\mathcal{M} = \{(0, 0), (1, 0), (1, 1)\}$, then we can draw the Voronoi diagram in Figure 1.

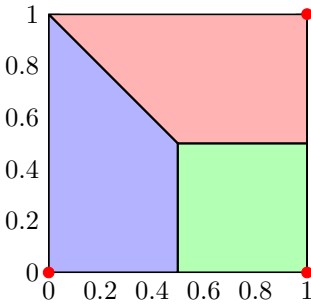

*Figure 1.* Voronoi diagram for $\mathcal{M} = \{(0, 0), (1, 0), (1, 1)\}$.

The weight $w_{\mathcal{M}}(y)$ exactly corresponds to the area of the region in the Voronoi diagram that corresponds to $y$. So for the alternatives in Figure 1, we have that $w_{\mathcal{M}}((0, 0)) = 0.375$, $w_{\mathcal{M}}((1, 0)) = 0.25$, and $w_{\mathcal{M}}((1, 1)) = 0.375$.

Now suppose $\mathcal{M}' = \mathcal{M} \cup \{(0.9, 1)\}$, i.e. $\mathcal{M}'$ contains a clone of the alternative $(1, 1)$. The Voronoi diagram of $\mathcal{M}'$ is shown in Figure 2.

We now have that $w_{\mathcal{M}'}((0, 0)) = 0.34$, $w_{\mathcal{M}'}((1, 0)) = 0.239875$, and $w_{\mathcal{M}'}((1, 1)) = 0.025$, $w_{\mathcal{M}'}((0.9, 1)) = 0.395125$. Therefore, the introduction of the approximate clone $x' = (0.9, 1)$ caused the weight of $x = (1, 1)$ to be split between $x$ and $x'$, and the weights of the other alternatives only changed by a small amount. Furthermore, because annotator preferences are continuous, we also have that $p_{\mathcal{D}}(x', y) \approx p_{\mathcal{D}}(x, y)$ for any alternative $y$. Therefore,

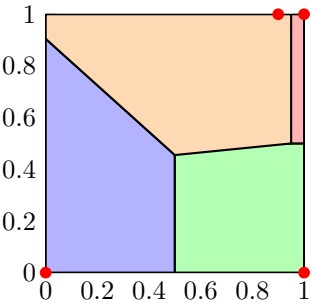

*Figure 2.* Diagram for $\mathcal{M} = \{(0, 0), (1, 0), (1, 1), (0.9, 1)\}$.

the introduction of the clone does not significantly change the weighted MLE objective function. In the proof of Theorem 4.2, we formally show this, and conclude that the weighted MLE reward function also does not significantly change.

The weighted MLE is not only robust to approximate clones, but also preserves many of the same interpretations as the standard MLE. For example, the weighted MLE and the standard MLE are equivalent whenever $\mathcal{M}$ is uniformly distributed across $\mathcal{S}$, i.e. when $w_{\mathcal{M}}(y) = \frac{1}{\mathcal{M}}$ for all $y \in \mathcal{M}$. In this case, Equation (2) is a scaled version of Equation (1), and therefore $\hat{r}^{\mathcal{D}} = \hat{\mathrm{r}}_{\mathrm{w}}^{\mathcal{D}}$. Therefore, the weighted MLE only differs from the standard MLE when the alternatives in $\mathcal{M}$ are not evenly distributed across $\mathcal{S}$.

To further understand this new algorithm, we discuss two additional perspectives on the weighted MLE and its relationship to the standard MLE.

**Relationship to Weighted Average Win Rate**

In Theorem 2.5 we established a strong connection between the MLE and the empirical average win-rate. Similarly, the weighted MLE has a close relationship with the *weighted average win rate* defined in Definition 4.3. Specifically, Theorem 4.4 shows that the weighted MLE is also an M-estimator solving a system of equations relating the weighted average win rate of $r$ to the empirical weighted average win rate in the data set $\mathcal{D}$.

**Definition 4.3.** For any dataset $\mathcal{D}$ over alternatives $\mathcal{M}$, the weighted average win rate of alternative $x \in \mathcal{M}$ is $\mathrm{wAWR}_{\mathcal{D}}(x) = \sum_{y \in \mathcal{M}} w_{\mathcal{M}}(y) p_{\mathcal{D}}(x \succ y)$.

**Theorem 4.4.** *For any dataset $\mathcal{D}$ over alternatives $\mathcal{M}$, the reward function $\hat{\mathrm{r}}_{\mathrm{w}}^{\mathcal{D}}$ from Equation (2) satisfies the system of equations*

$$\mathrm{wAWR}_{\mathcal{D}}(x) = \lambda \hat{\mathrm{r}}_{\mathrm{w}}^{\mathcal{D}}(x) + \widehat{\mathrm{wAWR}}(x) \quad \forall x \in \mathcal{M},$$

*where* $\widehat{\mathrm{wAWR}}(x) = \sum_{y \in \mathcal{M}} w_{\mathcal{M}}(y) \frac{e^{\hat{\mathrm{r}}_{\mathrm{w}}^{\mathcal{D}}(x)}}{e^{\hat{\mathrm{r}}_{\mathrm{w}}^{\mathcal{D}}(x)} + e^{\hat{\mathrm{r}}_{\mathrm{w}}^{\mathcal{D}}(y)}}.$

The proof of Theorem 4.4 can be found in Appendix E. Similar to the regularized MLE, a major consequence is that

the order in which the alternatives are ranked in the weighted MLE is the same as the order in which the alternatives are ranked by weighted average win rate.

**Corollary 4.5.** *For any dataset $\mathcal{D}$ over alternatives $\mathcal{M}$ and any $x, y \in \mathcal{M}$, $\hat{\mathrm{r}}_{\mathrm{w}}^{\mathcal{D}}(x) \geq \hat{\mathrm{r}}_{\mathrm{w}}^{\mathcal{D}}(y)$ if and only if $\mathrm{wAWR}_{\mathcal{D}}(x) \geq \mathrm{wAWR}_{\mathcal{D}}(y)$.*

### Interpretation as an MLE Approximation

One interpretation of the weighted MLE is that the function $f_{\mathcal{D}}(r)$ approximates what the regularized MLE objective would be if the dataset $\mathcal{M}$ contained every alternative in the entire alternative space $\mathcal{S}$. Approximating the regularized MLE objective for the entire alternative space $\mathcal{S}$ is a natural goal when the algorithm only is given information about a subset of alternatives $\mathcal{M} \subseteq \mathcal{S}$. The MLE objective for the entire alternative space $\mathcal{S}$ can be written as

$$f_{\mathcal{D}}^{\mathcal{S}}(r) = -\frac{1}{|\mathcal{S}|} \int_{y_1, y_2 \in \mathcal{S}} \mathcal{L}(y_1, y_2) dy_1 dy_2 + \frac{\lambda}{2} \int_{y \in \mathcal{S}} r(y)^2 dy \tag{3}$$

where the log likelihood term for the comparisons between $y_1$ and $y_2$ is defined as

$$\mathcal{L}(y_1, y_2) = p_{\mathcal{D}}(y_1 \succ y_2) \log \left( \frac{e^{r(y_1)}}{e^{r(y_1)} + e^{r(y_2)}} \right).$$

Theorem 4.6 (proven in Appendix H) shows how the weighted MLE objective can be written with the same structure as Equation (3) using using the $\mathrm{proj}_{\mathcal{M}}$ function to approximate the unknown quantities.

**Theorem 4.6.** *The weighted MLE objective function can be written in terms of $\mathcal{S}$ and the projection function $\mathrm{proj}_{\mathcal{M}}$ as:*

$$f_{\mathcal{D}}(r) = \frac{-1}{|\mathcal{S}|^2} \int_{y_1, y_2 \in \mathcal{S}} \overline{\mathcal{L}}(y_1, y_2) dy_1 dy_2 + \frac{\lambda}{2|\mathcal{S}|} \int_{y \in \mathcal{S}} \overline{r^2}(y) dy.$$

*where we define the estimated log likelihood of two alternatives $y_1, y_2 \in \mathcal{S}$ as*

$$\overline{\mathcal{L}}(y_1, y_2) := \frac{1}{|\mathrm{proj}_{\mathcal{M}}(y_1)| \cdot |\mathrm{proj}_{\mathcal{M}}(y_2)|}$$
$$\times \sum_{\substack{x_1 \in \mathrm{proj}_{\mathcal{M}}(y_1) \\ x_2 \in \mathrm{proj}_{\mathcal{M}}(y_2)}} p_{\mathcal{D}}(x_1 \succ x_2) \log \left( \frac{e^{r(x_1)}}{e^{r(x_1)} + e^{r(x_2)}} \right)$$

*and we define the estimated reward squared of an alternative $y \in \mathcal{S}$ as*

$$\overline{r^2}(y) := \frac{1}{|\mathrm{proj}_{\mathcal{M}}(y)|} \sum_{x \in \mathrm{proj}_{\mathcal{M}}(y)} r(x)^2.$$

For further intuition about Theorem 4.6, note that when $y_1$ and $y_2$ both have a unique closest alternative in $\mathcal{M}$ (i.e. $|\mathrm{proj}_{\mathcal{M}}(y_1)| = |\mathrm{proj}_{\mathcal{M}}(y_2)| = 1$), then

$$\overline{\mathcal{L}}(y_1, y_2) = p_{\mathcal{D}}(y_1' \succ y_2') \log \left( \frac{e^{r(y_1')}}{e^{r(y_1')} + e^{r(y_2')}} \right)$$

where $y_1'$ and $y_2'$ to be the unique elements of $\mathrm{proj}_{\mathcal{M}}(y_1)$ and $\mathrm{proj}_{\mathcal{M}}(y_2)$ respectively. In other words, in this case $\overline{\mathcal{L}}(y_1, y_2)$ is simply approximating $\mathcal{L}(y_1, y_2)$ using the closest alternatives in $\mathcal{M}$. Therefore, Theorem 4.6 shows that the weighted MLE also has a natural interpretation as an approximate MLE solution that only depends on $\mathcal{M}$ through the projection function $\mathrm{proj}_{\mathcal{M}}$.

## 5. Case Study

Although our contributions are primarily theoretical, we supplement our results with a synthetic case study that highlights an instance where the weighted MLE is more robust than the standard regularized MLE under diverse preferences. This case study moves our theory closer to practice in a few different ways. First, LLMs typically generate the responses seen by human annotators, and so our case study considers textual responses generated by the gpt-4o-mini model. Note that this means each alternative has only one data point, unlike our theoretical results where we assume sufficiently many comparisons for every pair of alternatives. This better represents what happens in practice when each pair of responses is newly generated by an LLM at the point when an annotator is asked to report a preference (Bai et al., 2022). LLMs have also been shown to be effective as implicit computational models of humans (Horton, 2023), and we therefore use an LLM as a stand-in for human annotators with diverse preferences rather than assuming humans make decisions using a BTL model.

In the case study, our goal is to train a reward function which evaluates answers to a single question: *'Describe Paris'*. We use OpenAI's gpt-4o-mini model both to generate textual descriptions of Paris and to simulate human annotators with diverse preferences. We consider a population with three types of annotators, each of which attach a different amount of importance to seeing the topics of food, art, and romance mentioned in a description of Paris. We then construct two preference datasets for this population, one which includes additional approximate clones ('Clones') and one which does not ('Original'). More details on the annotator population and the dataset generation process can be found in Appendix I.

We approximate both the standard MLE algorithm and the weighted MLE algorithm using neural networks. Each neural network takes as input a context vector and outputs a reward value. To generate the context vectors, we use OpenAI's text-embedding-3-small model to extract embedding vectors from the textual descriptions of Paris. More details on the neural network training process can be found in Appendix G. We run each algorithm on both datasets described in the previous paragraph and observe how the reward function output by the algorithm changes across datasets. To visualize the change in reward function, we evaluate each

reward function on all of the alternatives, and then plot the mean reward for three types of answers (food, art, and romance), with error bars corresponding to the sample standard deviations.

Figure 3 shows how the standard MLE algorithm performs on these two datasets. Training on dataset 'Original' leads to romance being the topic with the highest reward. However, training on dataset 'Clones' leads to art being the topic with the highest reward. The fact that the topic with the highest reward changes with the addition of clones highlights the lack of robustness of the MLE. Furthermore, we note that both datasets contain a relatively large amount of data, and therefore this noticeable change in the reward function cannot be attributed only to variance in the data generation or training processes.

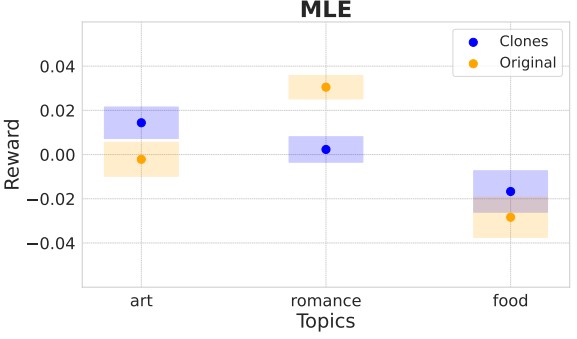

*Figure 3.* Results for the MLE: The yellow points show the average value of the MLE reward function for different topics when trained on dataset 'Original'. The blue points show the same but when trained on dataset 'Clones'. In the presence of clones, the rewards for both art and romance change significantly, showing that the MLE is not robust to clones.

Figure 4 shows the same results for the weighted MLE algorithm. Recall that the weighted MLE algorithm requires a choice of $\mathcal{S}$, which is the set of all possible alternatives. In the experiment shown in Figure 4, we chose $\mathcal{S}$ to be the unit cube in the high-dimensional context space. We also show that similar results hold for other choices of $\mathcal{S}$ in Appendix I. As shown in Figure 4, the average value for the weighted MLE of each of the different categories is roughly the same for both the dataset without clones and the dataset with clones. This shows that the weighted MLE is robust to the presence of clones, which aligns with the theoretical results of Section 4.

## 6. Discussion

In this section we discuss some limitations of our findings and outline potential directions for future research.

There is ample opportunity for further empirical research. Our case study is meant to highlight a specific instance

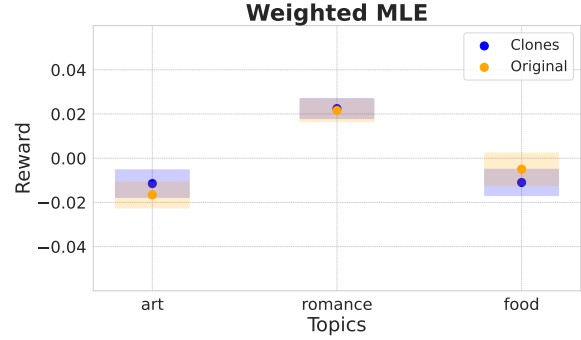

*Figure 4.* Results for the weighted MLE: The yellow points show the average value of the weighted MLE reward function for different topics when trained on dataset 'Original'. The blue points show the same but when trained on dataset 'Clones'. The rewards for the three topics do not change significantly, demonstrating the robustness of the weighted MLE.

where clones cause a problem for the standard RLHF methods, but does not imply any conclusions regarding how frequent or pervasive clones may be in practice. For example, we may not expect LLM-generated answers to the vanilla prompt 'describe Paris' to have such stark deviation into categories; rather, answers may be more balanced. The realized impact of approximate clones will also of course depend on the preferences of the annotator population. Further research could evaluate how often clones appear in practice and characterize the types of annotator populations which cause clones to be a problem.

In our theoretical results, we assume that we have sufficient comparisons between each pair of alternatives. This assumption may be unrealistic if the alternatives are answers generated by LLMs as in our case study, as then each response would only be involved in one comparison. The case study suggests a potential solution to this problem, which is to first cluster the original alternatives based on common features (or context) to form meta-alternatives. There could still exist approximate clones among these meta-alternatives; however, each meta-alternative would likely have a larger number of comparisons. One potential question for future work is to explore how different clustering schemes affect the robustness of both the MLE algorithm and the weighted MLE algorithm.

Even if each alternative is involved in multiple comparisons, it could be interesting to relax the assumption that we have sufficient comparisons between each pair of alternatives. As mentioned in Section 1, one choice for the alternatives in RLHF is the question/answer pairs. In this case, we would expect the dataset to only include comparisons between alternatives (question/answer pairs) where the question is the same. Although this would not exactly match the assumptions in our theory, we expect that similar theoretical results

would hold with regard to robustness to approximate clones.

Finally, we used a simple weighting scheme in Definition 4.1 to balance the objective function when the observed alternatives are not evenly distributed over the entire alternative space. However, this is not the unique weighting scheme that can achieve this desired result. One direction for future work is to experiment with different weighting schemes to see which perform the best in practice.

## Impact Statement

The focus of the paper is on AI alignment, a field whose ultimate goal is to make AI more beneficial. We acknowledge that, as with any work on AI alignment, there could be unforeseen negative consequences; further study is needed before our methods can be deployed.

## Acknowledgements

This work was partially supported by the National Science Foundation under grants IIS-2147187 and IIS-2229881; by the Office of Naval Research under grants N00014-24-1-2704 and N00014-25-1-2153; and by a grant from the Cooperative AI Foundation. Zhang and Schiffer were supported by an NSF Graduate Research Fellowship.

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

## A. Additional Related Work Details

In this section, we provide a more comprehensive comparison to several works that are especially related to ours. Like us, Xu et al. (2023) are concerned about the performance of current RLHF algorithms in the presence of duplicates. Like us, they show that there are simple models for human preferences under which the standard RLHF algorithms perform badly. Unlike us, their results are for a specific class of models they call dichotomy models. In such models, there are two types of messages and two types of individuals, and each type of individual has reward 1 for one type of message and reward 0 for the other. Their main results also focus on three-way comparisons, while our results deal with pairwise comparisons (which are standard in current RLHF algorithms).

We were inspired by Siththaranjan et al. (2023), who show that standard RLHF implicitly aggregates over hidden context according to Borda count. We build off their work to show that standard RLHF algorithms are not robust to approximate clones. Like Siththaranjan et al. (2023), we assume that humans have diverse preferences, but unlike them, we do not summarize these preferences by a hidden context. Rather, we directly model populations with different reward functions. Note that when we refer to 'context' in our paper, we are referring to the underlying context of alternatives, not of annotators as in Siththaranjan et al. (2023). Like us, Siththaranjan et al. (2023) give an impossibility result for RLHF algorithms in the presence of diverse preferences. They show that every RLHF algorithm fails to exactly recover the mean reward function for some population, while our result shows that every RLHF algorithm does arbitrarily badly at finding the mean reward function for some populations.

Chakraborty et al. (2024) also evaluate the efficacy of standard RLHF algorithms when there are diverse human preferences and give an impossibility result for when standard RLHF outputs a single reward function. However, their impossibility result is of a different form – they bound the gap between the optimal policy overall and the optimal policy for a subpopulation by the sum of total variation distances between preference distributions of subpopulations. By contrast, our impossibility result states that for any RLHF algorithm, there exists a population such that the distance between the RLHF algorithm output and the mean rewards of the population is arbitrarily large.

Finally, we also note that because we study RLHF with diverse populations, this work is inherently related to the study of pluralistic alignment of AI systems, see e.g. the work of Sorensen et al. (2024) and Anwar et al. (2024).

## B. Proof of Theorem 2.3

We will prove the desired result by contradiction.

We will show the desired result for any $C \geq \log^2(2)$ which will imply the desired result for all $C > 0$. For any $C \geq \log^2(2)$, define $\kappa := e^{12\sqrt{C}}$. Consider the following two populations, each of which consists of two types of annotators which are equally prevalent and two alternatives $a$ and $b$. The reward of each type of voter for each alternative in each population are shown in the two tables below.

**Population 1**

|        | Type 1 (50%) | Type 2 (50%) |
|--------|--------------|--------------|
| $r(a)$ | 0            | 0            |
| $r(b)$ | $\log(2)$    | $\log(2)$    |

**Population 2**

|        | Type 1 (50%)  | Type 2 (50%)                    |
|--------|---------------|---------------------------------|
| $r(a)$ | 0             | 0                               |
| $r(b)$ | $\log(\kappa)$ | $\log(\kappa + 4) - \log(2\kappa - 1)$ |

*Table 1.* Rewards for each annotator type in each population.

For population 1,

$$\Pr(a \succ b) = \frac{1}{2} \cdot \frac{e^0}{e^0 + e^{\log(2)}} + \frac{1}{2} \cdot \frac{e^0}{e^0 + e^{\log(2)}} = \frac{1}{3}.$$

For population 2,

$$\begin{aligned}
\Pr(a \succ b) &= \frac{1}{2} \cdot \frac{e^0}{e^0 + e^{\log(\kappa)}} + \frac{1}{2} \cdot \frac{e^0}{e^0 + e^{\log(\kappa+4) - \log(2\kappa-1)}} \\
&= \frac{1}{2} \cdot \frac{1}{1 + \kappa} + \frac{1}{2} \cdot \frac{1}{1 + \frac{\kappa+4}{2\kappa-1}} \\
&= \frac{1}{2} \cdot \frac{1}{1 + \kappa} + \frac{1}{2} \cdot \frac{2\kappa - 1}{3\kappa + 3}
\end{aligned}$$

$$= \frac{\frac{1}{2} + \frac{2\kappa - 1}{6}}{1 + \kappa}$$

$$= \frac{\frac{2\kappa + 2}{6}}{1 + \kappa}$$

$$= \frac{1}{3}.$$

Therefore, for both populations the probability that alternative $a$ is preferred to alternative $b$ is the same ($1/3$). This implies that based on query data, it is impossible to distinguish between these two populations. Now consider an arbitrary algorithm ALG that when given data such that $a$ is preferred to $b$ with probability $1/3$, ALG outputs reward function $r^{\mathcal{D}}$. We want to show that

$$\min_{\alpha \in \mathbb{R}} \sum_{x \in \mathcal{M}} \left( \frac{r_1^*(x) + r_2^*(x)}{2} - r^{\mathcal{D}}(x) - \alpha \right)^2 \leq C \tag{4}$$

cannot hold for both populations. If Equation (4) holds for population 1, then

$$\min_{\alpha \in \mathbb{R}} \left( \left( r^{\mathcal{D}}(a) + \alpha \right)^2 + \left( \log(2) - r^{\mathcal{D}}(b) - \alpha \right)^2 \right) \leq C,$$

which implies that there exists some $\alpha \in \mathbb{R}$ such that

$$\left| r^{\mathcal{D}}(a) + \alpha \right| \leq \sqrt{C}$$

and

$$\left| \log(2) - r^{\mathcal{D}}(b) - \alpha \right| \leq \sqrt{C}.$$

In order for the two equations above to both hold, we must have that

$$\left| r^{\mathcal{D}}(a) - r^{\mathcal{D}}(b) \right| \leq \log(2) + 2\sqrt{C}. \tag{5}$$

Similarly, if Equation (4) holds for population 2, then

$$\min_{\alpha \in \mathbb{R}} \left( \left( r^{\mathcal{D}}(a) + \alpha \right)^2 + \left( \frac{\log(\kappa) + \log(\kappa + 4) - \log(2\kappa - 1)}{2} - r^{\mathcal{D}}(b) - \alpha \right)^2 \right) \leq C$$

which implies that

$$\left| r^{\mathcal{D}}(a) + \alpha \right| \leq \sqrt{C}$$

and

$$\left| \frac{\log(\kappa) + \log(\kappa + 4) - \log(2\kappa - 1)}{2} - r^{\mathcal{D}}(b) - \alpha \right| \leq \sqrt{C}.$$

In order for the two above equations to hold, we must have that

$$\left| \frac{\log(\kappa) + \log(\kappa + 4) - \log(2\kappa - 1)}{2} + r^{\mathcal{D}}(a) - r^{\mathcal{D}}(b) \right| \leq 2\sqrt{C}.$$

In order for this equation to hold, we must have that

$$\begin{aligned}
\left| r^{\mathcal{D}}(a) - r^{\mathcal{D}}(b) \right| &\geq \frac{\log(\kappa) + \log(\kappa + 4) - \log(2\kappa - 1)}{2} - 2\sqrt{C} \\
&= \frac{\log(\kappa) + \log(\kappa + 4) - \log(\kappa - 1/2) - \log(2)}{2} - 2\sqrt{C} \\
&\geq \frac{\log(\kappa) - \log(2)}{2} - 2\sqrt{C} \\
&= \frac{12\sqrt{C} - \log(2)}{2} - 2\sqrt{C} \\
&\geq 4\sqrt{C} - \frac{\log(2)}{2}. \tag{6}
\end{aligned}$$

However, for $C \geq \log^2(2)$, Equations (5) and (6) cannot both hold, and therefore we have a contradiction.

# C. Adapting Independence of Clones

In this section, we provide further details and justification for our definition of independence of clones in the RLHF setting.

## C.1. Independence of Clones in Traditional Social Choice

In traditional social choice, independence of clones (Tideman, 1987) is a desirable characteristic of voting rules which intuitively states that the winner of an election remains the same when duplicates of candidates are added to the candidate pool. More specifically, classic voting theory considers settings with a set of $n$ voters (denoted $N$) and $m$ candidates (denoted $M$). Each voter then provides a full ranking over the $M$ candidates. A voting rule takes as input the set of rankings and outputs a single candidate. A subset $K \in M$ of candidates is a set of clones if no voter ranks any candidate in $M \setminus K$ between any two candidates in $K$. Finally, a voting rule is *Independent of Clones* if and only if the following two properties hold. First, a candidate in $M \setminus K$ is output by the voting rule if and only if that same candidate is output by the voting rule after eliminating any candidate in $K$. Second, a candidate in $K$ is output by the voting rule if and only if some other member of $K$ is also output by the voting rule after eliminating any candidate in $K$.

## C.2. Additional Justification for RLHF Definition

This section provides further justification for how we adapt the traditional independence of clones definition for RLHF. Here, we focus on how we develop a reasonable definition for exact independence of clones. In the next section, we will then explain why we further consider robustness to approximate clones.

Informally, we say that an RLHF algorithm satisfies exact independent of clones if adding new alternatives that are clones of existing alternatives does not change the reward function that is output by the RLHF algorithm. There are a few major differences between the definition of independence of clones in the traditional setting and in RLHF. First, while traditional independence of clones guarantees that the winning alternative does not change, the RLHF version of independence of clones instead guarantees that the reward function does not change, which is a stronger notion. This is because in traditional voting theory the focus is on the winning alternative, while in RLHF we care about the reward function over all of the alternatives. Second, the input to an RLHF algorithm consists of query results over pairs of alternatives, rather than full rankings from every voter. Therefore, the definition of a clone from traditional social choice does not carry over. Instead, we will define a clone in RLHF as a new alternative with the same *context* as an existing alternative, which implies that every voter has the same reward for the new alternative as for the existing one. Finally, it is generally assumed that an RLHF algorithm has access to noisy observations regarding human preferences, rather than the true rewards of each voter for each alternative. Due to randomness, it may be the case that two alternatives for which all voters have the exact same value may still look different in the dataset of query results given as input to the RLHF algorithm.

Therefore, we will say that an RLHF algorithm is independent of clones if *when the empirical pairwise win rates are the same as the true pairwise win rates*, then the reward function output by the algorithm is unchanged when a clone is added. Note that when the dataset contains sufficiently many queries, the empirical pairwise win rates will approximately equal the true pairwise win rates by the law of large numbers. Further note that it is not necessarily desirable that the reward function is unchanged when a clone is added if the empirical pairwise win rates are not close to the true win rates. This is because in this case, there is value in generating a larger dataset, and therefore it is no longer true that adding a clone adds no new information.

## C.3. Exact Independence of Clones

In this section, we formally present our definition of exact independence of clones and explain why our work primarily focuses on robustness to approximate clones. The formal definition of exact independence of clones is below.

**Definition C.1** (Exact Independence of Clones). An RLHF algorithm ALG satisfies independence of clones if the following holds. Consider a set of alternatives $[m + 1]$ such that the context of alternative $m$ is the same as the context of alternative $m + 1$. Let $\mathcal{D}_1$ and $\mathcal{D}_2$ be representative datasets over the alternative sets $[m]$ and $[m + 1]$ respectively. Let $r_1 = \mathrm{ALG}(\mathcal{D}_1)$, and $r_2 = \mathrm{ALG}(\mathcal{D}_2)$. Then $r_2(m + 1) = r_2(m)$ and for all $i \in [m]$, $r_1(i) = r_2(i)$.

We note that independence of clones as defined in Definition C.1 is a very weak guarantee, as two alternatives are only clones if their contexts are exactly equal and reward functions only need to remain unchanged when there is sufficient data. Even so, as a result of the equivalence between the regularized MLE for RLHF and Borda count as discussed in Section 2.2, the regularized MLE algorithm does not satisfy Definition C.1. We formally prove this result in Appendix D. However, because we have the context of the alternatives (as defined in Section 2), we can easily adapt the regularized MLE to satisfy independence of clones by a simple pre-processing step. Recall that two alternatives have the same context if and only if

they are clones. Therefore, we can combine the data of any two alternatives with the same context in order to remove clones. Note that the regularized MLE cannot be made to satisfy robustness of approximate clones with preprocessing, because Definition 3.1 must hold for any $\delta > 0$.

As further motivation for moving beyond exact independence of clones, RLHF queries often ask annotators to compare textual responses generated by LLMs, where it is unlikely that an exact response will be duplicated. Therefore, it is more realistic in RLHF to consider robustness to approximate clones. For example, an approximate clone of a textual response may substitute an adjective for its synonym, or use a slightly different grammar structure.

## D. Proof of Theorem 3.2

*Proof.* Let $\mathcal{M} = \{a, b, c\}$ and $\mathcal{M}' = \{a, b, c, c'\}$, where $\|c - c'\| = 0$. Note that $\mathcal{M}$ and $\mathcal{M}'$ differ only by $c'$, and $c'$ is an exact clone of an alternative $c \in \mathcal{M}$. Suppose that $\mathcal{D}$ is generated by querying a population which consists of three types of individuals, where each type is represented by a BTL model.

| Alternative | Type 1 (40%) | Type 2 (30%) | Type 3 (30%) |
|:-----------:|:------------:|:------------:|:------------:|
| r(a) | $\ln(100)$ | $\ln(10)$ | $\ln(1)$ |
| r(b) | $\ln(10)$ | $\ln(1)$ | $\ln(100)$ |
| r(c) | $\ln(1)$ | $\ln(100)$ | $\ln(10)$ |

*Table 2.* Representation of the population that generated $\mathcal{D}$. The proportions of each type are indicated in the header row, while the rewards associated with each type for every alternative are presented in the matrix.

Suppose further that $\mathcal{D}'$ is generated from the same population after cloning alternative $c$, and can be represented by the following:

| Alternative | Type 1 (40%) | Type 2 (30%) | Type 3 (30%) |
|:-----------:|:------------:|:------------:|:------------:|
| r(a) | $\ln(100)$ | $\ln(10)$ | $\ln(1)$ |
| r(b) | $\ln(10)$ | $\ln(1)$ | $\ln(100)$ |
| r(c) | $\ln(1)$ | $\ln(100)$ | $\ln(10)$ |
| r(c') | $\ln(1)$ | $\ln(100)$ | $\ln(10)$ |

Define the Borda Count of an alternative $x$ given the total set of alternatives $\mathcal{M}$ as $\mathrm{BC}(x, \mathcal{M}) = \sum_{y \in \mathcal{M}} p^*(x \succ y)$, and the Borda Count winner of $\mathcal{M}$ to be the $y \in \mathcal{M}$ with the highest Borda Count. Further let $\hat{r}^{\mathcal{D}}$ and $\hat{r}^{\mathcal{D}'}$ be the regularized MLE estimators for these two datasets. By Theorem 2.5, $\hat{r}^{\mathcal{D}}(x) > \hat{r}^{\mathcal{D}}(y)$ iff $\mathrm{BC}(x) > \mathrm{BC}(y)$, and similarly for $\hat{r}^{\mathcal{D}'}$.

To prove the theorem, it therefore suffices to show that the Borda Count winner of $\mathcal{M}'$ is not the same as the Borda Count winner in $\mathcal{M}$. To see why, observe that if the Borda Count winner of $\mathcal{M}'$ is $x$ and the Borda Count winner of $\mathcal{M}$ is $y$, then it must be that $\hat{r}^{\mathcal{D}}(x) < \hat{r}^{\mathcal{D}'}(x)$ or $\hat{r}^{\mathcal{D}}(y) > \hat{r}^{\mathcal{D}'}(y)$ (or both). If $\hat{r}^{\mathcal{D}}(x) < \hat{r}^{\mathcal{D}'}(x)$, then we can choose $\delta$ such that $\hat{r}^{\mathcal{D}'}(x) - \hat{r}^{\mathcal{D}}(x) > \delta > 0$. Then because $\|c - c'\| = 0$, there is no $\epsilon > 0$ for which $|\hat{r}^{\mathcal{D}}(x) - \hat{r}^{\mathcal{D}'}(x)| \leq \delta$, which implies that the regularized MLE is not robust to approximate clones.

It remains to be shown that the Borda Count winner of $\mathcal{M}'$ is not the same as the Borda Count winner in $\mathcal{M}$. Let $t_i$ represent the proportion of the population that is type $i$ and let $v(t_i, x)$ be the value of type $i$ for alternative $x$. For any two alternatives $x, y$ in $\mathcal{M}$, the win percentage of $x$ over $y$ is

$$p^*(x \succ y) = \sum_{i=1}^{3} t_i \cdot \frac{e^{v(t_i, x)}}{e^{v(t_i, x)} + e^{v(t_i, y)}}.$$

The following table gives $\mathrm{BC}(x, \mathcal{M})$ for every $x \in \mathcal{M}$. Note that in this table, alternative $a$ is the Borda Count winner.

Similarly, the following table gives $\mathrm{BC}(x, \mathcal{M}')$ for every $x \in \mathcal{M}'$. In this table, alternative $b$ is the Borda Count winner. We have therefore shown that the Borda Count winner of $\mathcal{M}'$ is not the same as the Borda Count winner in $\mathcal{M}$, which proves the theorem.

$\square$

|   | $p^*(x \succ a)$ | $p^*(x \succ b)$ | $p^*(x \succ c)$ | $\mathrm{BC}(x, \mathcal{M})$ |
|---|---|---|---|---|
| $a$ | 0.50 | 0.64 | 0.45 | 1.59 |
| $b$ | 0.36 | 0.50 | 0.64 | 1.50 |
| $c$ | 0.55 | 0.36 | 0.50 | 1.41 |

*Table 3.* Each row of this table represents an alternative $x$. The first three columns compute $p^*(x \succ y)$ for each alternative $y$. The last column gives the Borda Count of alternative $x$, which is the sum of the first three columns.

|   | $p^*(x \succ a)$ | $p^*(x \succ b)$ | $p^*(x \succ c)$ | $p^*(x \succ c')$ | $\mathrm{BC}(x, \mathcal{M})$ |
|---|---|---|---|---|---|
| $a$ | 0.50 | 0.64 | 0.45 | 0.45 | 2.04 |
| $b$ | 0.36 | 0.50 | 0.64 | 0.64 | 2.14 |
| $c$ | 0.55 | 0.36 | 0.50 | 0.50 | 1.91 |
| $c'$ | 0.55 | 0.36 | 0.50 | 0.50 | 1.91 |

## E. Proof of Theorem 4.4

We begin with the following lemma, which we will use multiple times throughout the proof.

**Lemma E.1.** *For an arbitrary set $\mathcal{T} \in \mathbb{R}^d$, let $w : \mathcal{T} \to \mathbb{R}_+$ such that $\sum_{y \in \mathcal{T}} w(y) = 1$. Let $\lambda \in \mathbb{R}^+$ and for any $x_1, x_2 \in \mathcal{T}$ let $p(x_1 \succ x_2) \in [0, 1]$. Define $\hat{r} := \arg\min_r f(r)$, where*

$$f(r) := -\sum_{x_1, x_2 \in \mathcal{T}} w(x_1)w(x_2)p(x_1 \succ x_2) \log\left(\frac{e^{r(x_1)}}{e^{r(x_1)} + e^{r(x_2)}}\right) + \frac{\lambda}{2} \sum_{y \in \mathcal{T}} w(y)r(y)^2.$$

*Then $f$ strongly convex with parameter $m = \lambda \min_{x \in \mathcal{T}} w(x)$. Therefore, $f$ has a unique global minimum $r^*$, and for any $r$,*

$$f(r) - f(r^*) \geq \frac{m}{2}\|r - r^*\|_2^2. \tag{7}$$

*Proof.* Note that the function $\log\left(\frac{e^{r(x_1)}}{e^{r(x_1)} + e^{r(x_2)}}\right)$ is strictly convex in $r(x_1)$ and $r(x_2)$ as shown in (Siththaranjan et al., 2023). Furthermore, for any $\lambda > 0$, because $w(x) > 0$ for all $x \in \mathcal{T}$, we have that $\frac{\lambda}{2}\sum_{x \in \mathcal{T}} w(x)r(x)^2$ is strongly convex in $r(x)$ for all $x \in \mathcal{T}$. Finally, adding a strongly convex function and a strictly convex function results in a strongly convex function. $\square$

*Proof of Theorem 4.4.* Recall that $\hat{r}_w^{\mathcal{D}} = \arg\min_r f_{\mathcal{D}}(r)$, where

$$f_{\mathcal{D}}(r) = -\sum_{x_1, x_2 \in \mathcal{M}} w_{\mathcal{M}}(x_1)w_{\mathcal{M}}(x_2)p_{\mathcal{D}}(x_1 \succ x_2) \log\left(\frac{e^{r(x_1)}}{e^{r(x_1)} + e^{r(x_2)}}\right) + \frac{\lambda}{2} \sum_{x \in \mathcal{M}} w_{\mathcal{M}}(x)r(x)^2.$$

By Lemma E.1, $\hat{r}_w^{\mathcal{D}}$ will be the solution to the equation setting the gradient of $f_{\mathcal{D}}(r)$ to 0. Define $\sigma(a) = \frac{e^a}{1+e^a}$. The gradient of $f_{\mathcal{D}}(r)$ is the following for $x \in \mathcal{M}$,

$$\frac{\partial f_{\mathcal{D}}(r)}{\partial r(x)}$$

$$= \lambda w_{\mathcal{M}}(x)r(x) - \sum_{y:y \neq x} w_{\mathcal{M}}(x)w_{\mathcal{M}}(y)\left(p_{\mathcal{D}}(x \succ y)\frac{e^{r(y)}}{e^{r(y)} + e^{r(x)}} + p_{\mathcal{D}}(y \succ x)\frac{-e^{r(x)}}{e^{r(y)} + e^{r(x)}}\right)$$

$$= \lambda w_{\mathcal{M}}(x)r(x) - \sum_{y:y \neq x} w_{\mathcal{M}}(x)w_{\mathcal{M}}(y)\left(p_{\mathcal{D}}(x \succ y)\left(1 - \frac{e^{r(x)}}{e^{r(y)} + e^{r(x)}}\right) - (1 - p_{\mathcal{D}}(x \succ y))\frac{e^{r(x)}}{e^{r(y)} + e^{r(x)}}\right)$$

$$= \lambda w_{\mathcal{M}}(x)r(x) + \sum_{y:y \neq x} w_{\mathcal{M}}(x)w_{\mathcal{M}}(y)\sigma(\hat{r}_w^{\mathcal{D}}(x) - \hat{r}_w^{\mathcal{D}}(y)) - w_{\mathcal{M}}(x)w_{\mathcal{M}}(y)p_{\mathcal{D}}(x \succ y). \tag{8}$$

Also note that

$$w_{\mathcal{M}}(y)\sigma(\hat{r}_w^{\mathcal{D}}(x) - \hat{r}_w^{\mathcal{D}}(x)) - w_{\mathcal{M}}(y)p_{\mathcal{D}}(x \succ x) = \frac{w_{\mathcal{M}}(y)}{2} - \frac{w_{\mathcal{M}}(y)}{2} = 0. \tag{9}$$

Dividing Equation (8) by $w_{\mathcal{M}}(x)$ and equating to 0 gives that

$$
\begin{aligned}
0 &= \frac{1}{w_{\mathcal{M}}(x)} \frac{\partial f_{\mathcal{D}}(r)}{\partial \hat{r}_{\mathrm{w}}^{\mathcal{D}}(x)} \\
&= \lambda \hat{r}_{\mathrm{w}}^{\mathcal{D}}(x) + \sum_{y:y \neq x} w_{\mathcal{M}}(y)\sigma(\hat{r}_{\mathrm{w}}^{\mathcal{D}}(x) - \hat{r}_{\mathrm{w}}^{\mathcal{D}}(y)) - w_{\mathcal{M}}(y)p_{\mathcal{D}}(x \succ y) \\
&= \lambda \hat{r}_{\mathrm{w}}^{\mathcal{D}}(x) + \sum_{y \in \mathcal{M}} w_{\mathcal{M}}(y)\sigma(\hat{r}_{\mathrm{w}}^{\mathcal{D}}(x) - \hat{r}_{\mathrm{w}}^{\mathcal{D}}(y)) - w_{\mathcal{M}}(y)p_{\mathcal{D}}(x \succ y) \qquad \text{Equation (9)} \\
&= \lambda \hat{r}_{\mathrm{w}}^{\mathcal{D}}(x) + \sum_{y \in \mathcal{M}} w_{\mathcal{M}}(y)\sigma(\hat{r}_{\mathrm{w}}^{\mathcal{D}}(x) - \hat{r}_{\mathrm{w}}^{\mathcal{D}}(y)) - \sum_{y \in \mathcal{M}} w_{\mathcal{M}}(y)p_{\mathcal{D}}(x \succ y) \\
&= \lambda \hat{r}_{\mathrm{w}}^{\mathcal{D}}(x) + \sum_{y \in \mathcal{M}} w_{\mathcal{M}}(y)\sigma(\hat{r}_{\mathrm{w}}^{\mathcal{D}}(x) - \hat{r}_{\mathrm{w}}^{\mathcal{D}}(y)) - \mathrm{wAWR}_{\mathcal{D}}(x)
\end{aligned}
$$

Rearranging the sides, we have that

$$
\mathrm{wAWR}_{\mathcal{D}}(x) = \lambda \hat{r}_{\mathrm{w}}^{\mathcal{D}}(x) + \sum_{y \in \mathcal{M}} w_{\mathcal{M}}(y)\sigma(\hat{r}_{\mathrm{w}}^{\mathcal{D}}(x) - \hat{r}_{\mathrm{w}}^{\mathcal{D}}(y))
$$

as desired.

$\square$

## F. Proof of Theorem 2.5

*Proof.* This result follows exactly the same steps as in the proof of Theorem 4.4, except substituting $w_{\mathcal{M}}(x) = 1$ for all $x \in \mathcal{M}$. $\square$

## G. Proof of Theorem 4.2

We begin with the following lemma.

**Lemma G.1.** *For an arbitrary set $\mathcal{T} \in \mathbb{R}^d$, let $w : \mathcal{T} \to \mathbb{R}_+$ such that $\sum_{y \in \mathcal{T}} w(y) = 1$. Let $\lambda \in \mathbb{R}^+$ and for any $x_1, x_2 \in \mathcal{T}$ let $p(x_1 \succ x_2) \in [0, 1]$. Define $\hat{r} := \arg\min_r f(r)$, where*

$$
f(r) := -\sum_{x_1, x_2 \in \mathcal{T}} w(x_1)w(x_2)p(x_1 \succ x_2) \log\left(\frac{e^{r(x_1)}}{e^{r(x_1)} + e^{r(x_2)}}\right) + \frac{\lambda}{2}\sum_{y \in \mathcal{T}} w(y)r(y)^2.
$$

*Then*

$$
0 < f(\hat{r}) \leq 1 + \frac{\lambda}{2}
$$

*and for all $y \in \mathcal{T}$*

$$
|\hat{r}(y)| \leq \sqrt{\frac{2 + \lambda}{\lambda w(y)}}.
$$

*Proof.* If $r(y) = 1$ for all $y$ then $f(r) \leq 1 + \frac{\lambda}{2}$. This implies that $f(\hat{r}) \leq 1 + \frac{\lambda}{2}$. Furthermore, note that no finite values of $r$ can make $f(r) = 0$ due to the log terms, and therefore $f(\hat{r}) > 0$.

Finally, using that $f(\hat{r}) \leq 1 + \frac{\lambda}{2}$, it must be the case that $\frac{\lambda}{2}w(y)\hat{r}(y)^2 \leq 1 + \frac{\lambda}{2}$. Rearranging terms gives the desired bound on $|\hat{r}(y)|$. $\square$

Now we are ready to prove Theorem 4.2.

*Proof of Theorem 4.2.* For any set of alternatives $\mathcal{M}$ and any $\delta > 0$, we will define $\epsilon := c \cdot \delta^2$, where $c$ is a constant relative to $\delta$ that depends on $\mathcal{M}$ which we will defer the definition of until after Equations (10) and (11). Now consider any $x'$ such that $\|x' - x\|_2 \leq \epsilon$ for some $x \in \mathcal{M}$. Define $\mathcal{M}' = \mathcal{M} \cup \{x'\}$, and suppose $\mathcal{D}$ and $\mathcal{D}'$ are representative datasets on $\mathcal{M}$ and $\mathcal{M}'$ respectively. Our goal is to bound the difference between $\hat{r}_{\mathrm{w}}^{\mathcal{D}}$ and $\hat{r}_{\mathrm{w}}^{\mathcal{D}'}$, which are defined as follows.

$\hat{r}_w^{\mathcal{D}} = \arg\min_r f_{\mathcal{D}}(r)$, where

$$f_{\mathcal{D}}(r) = -\sum_{x_1,x_2 \in \mathcal{M}} w_{\mathcal{M}}(x_1)w_{\mathcal{M}}(x_2)p_{\mathcal{D}}(x_1 \succ x_2)\log\left(\frac{e^{r(x_1)}}{e^{r(x_1)} + e^{r(x_2)}}\right) + \frac{\lambda}{2}\sum_{x \in \mathcal{M}} w_{\mathcal{M}}(x)r(x)^2.$$

$\hat{r}_w^{\mathcal{D}'} = \arg\min_r f_{\mathcal{D}'}(r)$, where

$$f_{\mathcal{D}'}(r) = -\sum_{x_1,x_2 \in \mathcal{M}'} w_{\mathcal{M}'}(x_1)w_{\mathcal{M}'}(x_2)p_{\mathcal{D}'}(x_1 \succ x_2)\log\left(\frac{e^{r(x_1)}}{e^{r(x_1)} + e^{r(x_2)}}\right) + \frac{\lambda}{2}\sum_{x \in \mathcal{M}'} w_{\mathcal{M}'}(x)r(x)^2.$$

The objective functions $f_{\mathcal{D}}(r)$ and $f_{\mathcal{D}'}(r)$ differ in three ways. First, they have different weights ($w_{\mathcal{M}}$ versus $w_{\mathcal{M}'}$). Second, they use different sets of alternatives ($\mathcal{M}$ versus $\mathcal{M}'$). Third, they have different comparison probability functions ($p_{\mathcal{D}}$ versus $p_{\mathcal{D}'}$). In order to compare $\hat{r}_w^{\mathcal{D}}$ to $\hat{r}_w^{\mathcal{D}'}$, we will first change the weight functions to match, then change the sets of alternatives to match, and then finally change the comparison probability functions to match. Therefore, we will consider two intermediate reward functions $r_1$ and $r_2$ that are the optimal reward functions for objectives $f_1$ and $f_2$ respectively. Informally, $f_1$ can be viewed as being the same as $f_{\mathcal{D}}$ except that it uses different weights that correspond to the weights from $f_{\mathcal{D}'}$. Similarly, $f_2$ can be viewed as being the same as $f_{\mathcal{D}'}$ except it uses a different comparison probability function that corresponds to the comparison probability function from $f_{\mathcal{D}}$. Finally, $f_1$ and $f_2$ can be viewed as being the same except that they use the sets $\mathcal{M}$ and $\mathcal{M}'$ respectively.

Next we formally define $r_1, f_1$ and $r_2, f_2$. Define the function $w_1 : \mathcal{M} \to \mathbb{R}$ as $w_1(y) = w_{\mathcal{M}'}(y)$ for all $y \in \mathcal{M} \setminus \{x\}$ and $w_1(x) = w_{\mathcal{M}'}(x) + w_{\mathcal{M}'}(x')$. In other words, $w_1$ is the same as $w_{\mathcal{M}'}$ except it allocates all of the probability of both $x$ and $x'$ to alternative $x$. Then we can define $r_1$ as

$r_1 := \arg\min_r f_1(r)$, where

$$f_1(r) = -\sum_{x_1,x_2 \in \mathcal{M}} w_1(x_1)w_1(x_2)p_{\mathcal{D}}(x_1 \succ x_2)\log\left(\frac{e^{r(x_1)}}{e^{r(x_1)} + e^{r(x_2)}}\right) + \frac{\lambda}{2}\sum_{z \in \mathcal{M}} w_1(z)r(z)^2.$$

Define $p_2(x_1 \succ x_2) = p_{\mathcal{D}}(x_1 \succ x_2)$ for all $x_1, x_2 \in \mathcal{M}$, and define $p_2(y \succ x') = p_{\mathcal{D}}(y \succ x)$ and $p_2(x' \succ y) = p_{\mathcal{D}}(x \succ y)$ for all $y \in \mathcal{M}$. In other words, $p_2$ is the same as $p_{\mathcal{D}}$ except that every comparison involving $x'$ has the same probability as the corresponding comparison involving $x$. Then we can define $r_2$ as

$r_2 := \arg\min_r f_2(r)$, where

$$f_2(r) = -\sum_{x_1,x_2 \in \mathcal{M}'} w_{\mathcal{M}'}(x_1)w_{\mathcal{M}'}(x_2)p_2(x_1 \succ x_2)\log\left(\frac{e^{r(x_1)}}{e^{r(x_1)} + e^{r(x_2)}}\right) + \frac{\lambda}{2}\sum_{z \in \mathcal{M}'} w_{\mathcal{M}'}(z)r(z)^2$$

Lemma G.2 bounds the difference between $\hat{r}_w^{\mathcal{D}}$ and $r_1$, Lemma G.3 bounds the difference between $r_1$ and $r_2$, and Lemma G.4 bounds the difference between $r_2$ and $\hat{r}_w^{\mathcal{D}'}$. Using the triangle inequality, we can combine these three lemmas to get that for any $y \in \mathcal{M}$,

$$\begin{aligned}
|\hat{r}_w^{\mathcal{D}}(y) - \hat{r}_w^{\mathcal{D}'}(y)| &= |\hat{r}_w^{\mathcal{D}}(y) - r_1(y) + r_1(y) - r_2(y) + r_2(y) - \hat{r}_w^{\mathcal{D}'}(y)| \\
&\leq |\hat{r}_w^{\mathcal{D}}(y) - r_1(y)| + |r_1(y) - r_2(y)| + |r_2(y) - \hat{r}_w^{\mathcal{D}'}(y)| \\
&\leq O(\sqrt{\epsilon}) \qquad\qquad\qquad\qquad\qquad\qquad\text{[Lemmas G.2, G.3, G.4]} \qquad (10)
\end{aligned}$$

and that

$$\begin{aligned}
|\hat{r}_w^{\mathcal{D}'}(x) - \hat{r}_w^{\mathcal{D}'}(x')| &= |\hat{r}_w^{\mathcal{D}'}(x) - r_2(x') + r_2(x') - \hat{r}_w^{\mathcal{D}'}(x')| \\
&\leq |\hat{r}_w^{\mathcal{D}'}(x) - r_2(x')| + |r_2(x') - \hat{r}_w^{\mathcal{D}'}(x')| \\
&\leq |\hat{r}_w^{\mathcal{D}'}(x) - r_2(x)| + |r_2(x') - \hat{r}_w^{\mathcal{D}'}(x')| \qquad\text{[Lemma G.3]} \\
&\leq O(\sqrt{\epsilon}). \qquad\qquad\qquad\qquad\qquad\qquad\qquad\text{[Lemma G.4]} \qquad (11)
\end{aligned}$$

If we define $c$ in the definition of $\epsilon$ such that the $O(\sqrt{\epsilon})$ from the two above equations is bounded by $\delta$, then this exactly shows the desired result of Theorem 4.2.

The rest of the proof will focus on stating and proving Lemmas G.2, G.3, and G.4.

**Lemma G.2.** *For all $y \in \mathcal{M}$,*

$$|\hat{r}_w^{\mathcal{D}}(y) - r_1(y)| \leq O(\sqrt{\epsilon}). \tag{12}$$

*Proof.* First, we note that we can choose $\epsilon$ to be sufficiently small and assume WLOG that $x$ is the closest alternative in $\mathcal{M}$ to $x'$. This implies that

$$w_{\mathcal{M}}(x) \leq w_1(x) \leq w_{\mathcal{M}}(x) + O(\epsilon),$$

and that for every $y \in \mathcal{M} \setminus \{x\}$

$$w_{\mathcal{M}}(y) - O(\epsilon) \leq w_1(y) \leq w_{\mathcal{M}}(y).$$

The previous two equations together imply that for any $y \in \mathcal{M}$,

$$|w_{\mathcal{M}}(y) - w_1(y)| \leq O(\epsilon). \tag{13}$$

Note that we also have that for any $a, b \in \mathbb{R}$,

$$\log\left(\frac{e^a}{e^a + e^b}\right) \geq -(2 + |a| + |b|). \tag{14}$$

Therefore, for any $r$ satisfying $|r(y)| \leq C$ for all $y \in \mathcal{M}$ for some constant $C$, we have that

$$
\begin{aligned}
&f_1(r) - f_{\mathcal{D}}(r) \\
&\geq - \sum_{x_1, x_2 \in \mathcal{M}} (w_1(x_1)w_1(x_2) - w_{\mathcal{M}}(x_1)w_{\mathcal{M}}(x_2))p_{\mathcal{D}}(x_1 \succ x_2) \log\left(\frac{e^{r(x_1)}}{e^{r(x_1)} + e^{r(x_2)}}\right) + \frac{\lambda}{2} \sum_{y \in \mathcal{M}} (w_1(y) - w_{\mathcal{M}}(y))r(x)^2 \\
&\geq - \sum_{x_1, x_2 \in \mathcal{M}} (-O(\epsilon)w_1(x_1) - O(\epsilon)w_1(x_2))p_{\mathcal{D}}(x_1 \succ x_2) \log\left(\frac{e^{r(x_1)}}{e^{r(x_1)} + e^{r(x_2)}}\right) + \frac{\lambda}{2} \sum_{y \in \mathcal{M}} -O(\epsilon)r(x)^2 \\
&= - \sum_{x_1, x_2 \in \mathcal{M}} O(\epsilon)(w_1(x_1) + w_1(x_2))p_{\mathcal{D}}(x_1 \succ x_2)(2 + |r(x_1)| + |r(x_2)|) + \frac{\lambda}{2} \sum_{y \in \mathcal{M}} -O(\epsilon)r(x)^2 \\
&= -O(\epsilon). \tag{15}
\end{aligned}
$$

Similarly, for any $r$ satisfying $|r(y)| \leq C$ for all $y \in \mathcal{M}$ and some constant $C$, we have that

$$f_1(r) - f_{\mathcal{D}}(r) \leq O(\epsilon). \tag{16}$$

By Lemma G.1, we have for all $y \in \mathcal{M}$ that $|r_w^{\mathcal{D}}(y)| \leq \sqrt{\frac{2+\lambda}{\lambda w_{\mathcal{M}}(y)}}$ and $|r_1(y)| \leq \sqrt{\frac{2+\lambda}{\lambda w_1(y)}}$. Therefore, by Equations (15) and (16), we have that for constants $C_1, C_2$,

$$f_1(r_1) - f_{\mathcal{D}}(r_1) \geq -C_1\epsilon \tag{17}$$

and

$$f_{\mathcal{D}}(r_w^{\mathcal{D}}) - f_1(r_w^{\mathcal{D}}) \geq -C_2\epsilon. \tag{18}$$

Now we will show that

$$f_{\mathcal{D}}(r_1) - f_{\mathcal{D}}(\hat{r}_w^{\mathcal{D}}) \leq (C_1 + C_2)\epsilon. \tag{19}$$

Suppose Equation (19) does not hold. Then

$$
\begin{aligned}
f_1(r_1) - f_1(\hat{r}_w^{\mathcal{D}}) &= (f_1(r_1) - f_{\mathcal{D}}(r_1)) + \left(f_{\mathcal{D}}(r_1) - f_{\mathcal{D}}(\hat{r}_w^{\mathcal{D}})\right) + \left(f_{\mathcal{D}}(\hat{r}_w^{\mathcal{D}}) - f_1(\hat{r}_w^{\mathcal{D}})\right) \\
&> -C_1\epsilon + (C_1 + C_2)\epsilon - C_2\epsilon \qquad\qquad\qquad \text{[Eqs (18) and (17)]} \\
&= 0. \tag{20}
\end{aligned}
$$

This is a contradiction with the definition of $r_1$, and therefore Equation (19) must hold.

We will now use the contrapositive of Equation (19) to prove the desired result. Lemma E.1 implies that for any $r$ such that there exists a $y \in \mathcal{M}$ where $|r(y) - \hat{r}_w^{\mathcal{D}}(y)| > \sqrt{\frac{(C_1+C_2)\epsilon}{\lambda \min_{z \in \mathcal{M}} w_{\mathcal{M}}(z)}}$, we must have

$$f_{\mathcal{D}}(r) - f_{\mathcal{D}}(\hat{r}_w^{\mathcal{D}}) > (C_1 + C_2)\epsilon.$$

This combined with Equation (19) implies that for all $y$, $|r(y) - \hat{r}_w^{\mathcal{D}}(y)| \leq \sqrt{\frac{(C_1+C_2)\epsilon}{\lambda \min_{z \in \mathcal{M}} w_{\mathcal{M}}(z)}} = O(\sqrt{\epsilon})$, which is the desired result. $\qquad\square$

**Lemma G.3.** *For all $y \in \mathcal{M}$, $r_1(y) = r_2(y)$. Furthermore, $r_2(x') = r_2(x)$.*

*Proof.* Define $\sigma(a) = \frac{e^a}{1+e^a}$. Theorem 4.4 implies that $r_1$ is the solution to the set of equations

$$\sum_{y \in \mathcal{M}} w_1(y) p_{\mathcal{D}}(z \succ y) = \lambda r_1(z) + \sum_{y \in \mathcal{M}} w_1(y)\sigma(r_1(z) - r_1(y)) \quad \forall z \in \mathcal{M}. \tag{21}$$

which by definition of $p_2$ is equivalent to being the solution to this set of equations:

$$\sum_{y \in \mathcal{M}} w_1(y) p_2(z \succ y) = \lambda r_1(z) + \sum_{y \in \mathcal{M}} w_1(y)\sigma(r_1(z) - r_1(y)) \quad \forall z \in \mathcal{M}. \tag{22}$$

Similarly, $r_2$ is the solution to the set of equations

$$\sum_{y \in \mathcal{M}'} w_{\mathcal{M}'}(y) p_2(z \succ y) = \lambda r_2(z) + \sum_{y \in \mathcal{M}'} w_{\mathcal{M}'}(y)\sigma(r_2(z) - r_2(y)) \quad \forall z \in \mathcal{M}'. \tag{23}$$

Because $p_2(x' \succ y) = p_2(x \succ y)$, we see that the LHS of Equation (23) is the same for $x$ and for $x'$. Therefore, $r_2(x)$ and $r_2(x')$ satisfy the same equation (which only has one solution), and therefore

$$r_2(x) = r_2(x').$$

Now we will show that this implies that $r_1(y) = r_2(y)$ for all $y \in \mathcal{M}$. By definition of $w_1$, for all $z$ we have that

$$\begin{aligned}
\sum_{y \in \mathcal{M}} w_1(y) p_2(z \succ y) &= w_1(x) p_2(z \succ x) + \sum_{y \in \mathcal{M} \setminus x} w_1(y) p_2(z \succ y) \\
&= (w_{\mathcal{M}'}(x) + w_{\mathcal{M}'}(x')) p_2(z \succ x) + \sum_{y \in \mathcal{M} \setminus x} w_{\mathcal{M}'}(y) p_2(z \succ y) \\
&= \sum_{y \in \mathcal{M}'} w_{\mathcal{M}'}(y) p_2(z \succ y). \tag{24}
\end{aligned}$$

Because $r_2(x) = r_2(x')$, we also have that for all $z$,

$$\begin{aligned}
&\lambda r_2(z) + \sum_{y \in \mathcal{M}'} w_{\mathcal{M}'}(y)\sigma(r_2(z) - r_2(y)) \\
&= \lambda r_2(z) + w_{\mathcal{M}'}(x)\sigma(r_2(z) - r_2(x)) + w_{\mathcal{M}'}(x')\sigma(r_2(z) - r_2(x')) + \sum_{y \in \mathcal{M}' \setminus \{x,x'\}} w_{\mathcal{M}'}(y)\sigma(r_2(z) - r_2(y)) \\
&= \lambda r_2(z) + w_{\mathcal{M}'}(x)\sigma(r_2(z) - r_2(x)) + w_{\mathcal{M}'}(x')\sigma(r_2(z) - r_2(x)) + \sum_{y \in \mathcal{M}' \setminus \{x,x'\}} w_{\mathcal{M}'}(y)\sigma(r_2(z) - r_2(y)) \\
&= \lambda r_2(z) + (w_{\mathcal{M}'}(x) + w_{\mathcal{M}'}(x'))\sigma(r_2(z) - r_2(x)) + \sum_{y \in \mathcal{M}' \setminus \{x,x'\}} w_{\mathcal{M}'}(y)\sigma(r_2(z) - r_2(y)) \\
&= \lambda r_2(z) + w_1(x)\sigma(r_2(z) - r_2(x)) + \sum_{y \in \mathcal{M} \setminus \{x\}} w_1(y)\sigma(r_2(z) - r_2(y))
\end{aligned}$$

$$= \lambda r_2(z) + \sum_{y \in \mathcal{M}} w_1(y) \sigma(r_2(z) - r_2(y)). \tag{25}$$

Combining Equations (23), (24), and (25) gives that $r_2$ satisfies the equation

$$\sum_{y \in \mathcal{M}} w_1(y) p_2(z \succ y) = \lambda r_2(z) + \sum_{y \in \mathcal{M}} w_1(y) \sigma(r_2(z) - r_2(y)) \quad \forall z \in \mathcal{M}.$$

This means that $r_2$ and $r_1$ are solutions to the same set of equations, and therefore $r_2(y) = r_1(y)$ for all $y \in \mathcal{M}$. □

Finally, we can relate $r_2$ to $\hat{r}_{\mathrm{w}}^{\mathcal{D}'}$.

**Lemma G.4.** *For all $y \in \mathcal{M}'$, $|\hat{r}_{\mathrm{w}}^{\mathcal{D}'}(y) - r_2(y)| \leq O(\sqrt{\epsilon})$.*

*Proof.* Because $\mathcal{D}$ and $\mathcal{D}'$ are both representative datasets, by definition of $p_2$, for all $x_1, x_2 \in \mathcal{M}$ we have that $p_2(x_1 \succ x_2) = p_{\mathcal{D}}(x_1 \succ x_2) = p_{\mathcal{D}'}(x_1 \succ x_2)$. Therefore, only way in which $f_2(r)$ and $f_{\mathcal{D}'}(r)$ differ is that $p_2(y \succ x') = p_{\mathcal{D}}(y \succ x) \neq p_{\mathcal{D}'}(y \succ x')$ for all $y \in \mathcal{M}$ (and same for $p_2(x' \succ y)$). Because $\mathcal{D}$ and $\mathcal{D}'$ are both representative data sets, the preferences are Lipschitz continuous, and the BTL model is Lipschitz continuous, we have for some constant $C_1 > 0$ that

$$|p_{\mathcal{D}}(x \succ y) - p_{\mathcal{D}'}(x' \succ y)| = |p_{\mathcal{D}'}(x \succ y) - p_{\mathcal{D}'}(x' \succ y)| \leq C_1 \epsilon.$$

and

$$|p_{\mathcal{D}}(y \succ x) - p_{\mathcal{D}'}(y \succ x')| = |p_{\mathcal{D}'}(y \succ x) - p_{\mathcal{D}'}(y \succ x')| \leq C_1 \epsilon.$$

Next, recall that by Lemma G.1, $r_w^{\mathcal{D}'}$ and $r_2$ both are bounded by $\sqrt{\frac{2+\lambda}{\lambda w_{\mathcal{M}'}}}$. For any $r$ satisfying $|r(y)| \leq \sqrt{\frac{2+\lambda}{\lambda w_{\mathcal{M}'}}}$, we also have that for some constant $C_2 > 0$,

$$
\begin{aligned}
& f_{\mathcal{D}'}(r) - f_2(r) \\
&= -\sum_{y \in \mathcal{M}} w_{\mathcal{M}'}(x') w_{\mathcal{M}'}(y) \left( (p_{\mathcal{D}'}(x' \succ y) - p_2(x' \succ y)) \sigma(r(x') - r(y)) + (p_{\mathcal{D}'}(y \succ x') - p_2(y \succ x')) \sigma\left(r(y) - r(x')\right) \right) \\
&= -\sum_{y \in \mathcal{M}} w_{\mathcal{M}'}(x') w_{\mathcal{M}'}(y) \left( (p_{\mathcal{D}'}(x' \succ y) - p_{\mathcal{D}}(x \succ y)) \sigma(r(x') - r(y)) + (p_{\mathcal{D}'}(y \succ x') - p_{\mathcal{D}}(y \succ x)) \sigma\left(r(y) - r(x')\right) \right) \\
&\leq -C_1 \epsilon \sum_{y \in \mathcal{M}} w_{\mathcal{M}'}(x') w_{\mathcal{M}'}(y) \left( \sigma(r(x') - r(y)) + \sigma\left(r(y) - r(x')\right) \right) \\
&\leq C_1 \epsilon \left( \sum_{y \in \mathcal{M}} w_{\mathcal{M}'}(x') w_{\mathcal{M}'}(y) \left(2 + |r(y)| + |r(x')| + 2 + |r(y)| + |r(x')|\right) \right) \qquad \text{[Equation (14)]} \\
&\leq C_1 \epsilon \left( \sum_{y \in \mathcal{M}} w_{\mathcal{M}'}(x') w_{\mathcal{M}'}(y) \left(4 + 4\sqrt{\frac{2+\lambda}{\lambda w_{\mathcal{M}'}}}\right) \right) \\
&\leq C_2 \epsilon. \tag{26}
\end{aligned}
$$

By the same logic we also have that

$$f_2(r) - f_{\mathcal{D}'}(r) \leq C_2 \epsilon. \tag{27}$$

Note that Equations (26) and (27) hold for $r = r_w^{\mathcal{D}'}$ and $r = r_2$ by Lemma G.1, Next we will show that

$$f_2(r_w^{\mathcal{D}'}) - f_2(r_2) \leq 2C_2 \epsilon. \tag{28}$$

Assume Equation (28) does not hold. Then we have that

$$
\begin{aligned}
f_{\mathcal{D}'}(r_w^{\mathcal{D}'}) - f_{\mathcal{D}'}(r_2) &= (f_{\mathcal{D}'}(r_w^{\mathcal{D}'}) - f_2(r_w^{\mathcal{D}'})) + (f_2(r_w^{\mathcal{D}'}) - f_2(r_2)) + (f_2(r_2) - f_{\mathcal{D}'}(r_2)) \\
&> -C_2 \epsilon + 2C_2 \epsilon - C_2 \epsilon \\
&= 0.
\end{aligned}
$$

However, this is a contradiction because by definition of $r_w^{\mathcal{D}'}$, we must have that $f_{\mathcal{D}'}(r_w^{\mathcal{D}'}) - f_{\mathcal{D}'}(r_1) \leq 0$. Therefore, we have shown that Equation (28) holds.

Lemma E.1 implies that for any $r$ such that there exists a $y \in \mathcal{M}'$ such that $|r(y) - r_2(y)| > \sqrt{\frac{2C_2\epsilon}{\lambda \min_{z \in \mathcal{M}} w_{\mathcal{M}}(z)}}$, we must have that

$$f_2(r) - f_2(r_2) > 2C_2\epsilon.$$

The contrapositive of the previous statement combined with Equation (28) implies that for all $y$, we must have

$$|r_w^{\mathcal{D}'}(y) - r_2(y)| \leq \sqrt{\frac{2C_2\epsilon}{\lambda \min_{z \in \mathcal{M}} w_{\mathcal{M}}(z)}} = O(\sqrt{\epsilon}).$$

$\square$

$\square$

# H. Proof of Theorem 4.6
*Proof.*

$$f_{\mathcal{D}}(r) = -\sum_{x_1, x_2 \in \mathcal{M}} \left( w_{\mathcal{M}}(x_1) w_{\mathcal{M}}(x_2) p_{\mathcal{D}}(x_1 \succ x_2) \times \log\left(\frac{e^{r(x_1)}}{e^{r(x_1)} + e^{r(x_2)}}\right)\right) + \frac{\lambda}{2} \sum_{x \in \mathcal{M}} w_{\mathcal{M}}(x) r(x)^2$$

$$= -\sum_{x_1, x_2 \in \mathcal{M}} \left( \left(\frac{1}{|\mathcal{S}|} \int_{y_1 \in \mathcal{S}} \frac{1_{x_1 \in \text{proj}_{\mathcal{M}}(y_1)}}{|\text{proj}_{\mathcal{M}}(y_1)|} dy_1\right) \left(\frac{1}{|\mathcal{S}|} \int_{y_2 \in \mathcal{S}} \frac{1_{x_2 \in \text{proj}_{\mathcal{M}}(y_2)}}{|\text{proj}_{\mathcal{M}}(y_2)|} dy_2\right) p_{\mathcal{D}}(x_1 \succ x_2) \times \log\left(\frac{e^{r(x_1)}}{e^{r(x_1)} + e^{r(x_2)}}\right)\right)$$

$$+ \frac{\lambda}{2} \sum_{x \in \mathcal{M}} \left(\frac{1}{|\mathcal{S}|} \int_{y \in \mathcal{S}} \frac{1_{x \in \text{proj}_{\mathcal{M}}(y)}}{|\text{proj}_{\mathcal{M}}(y)|} dy\right) r(x)^2$$

$$= -\frac{1}{|\mathcal{S}|^2} \int_{y_1, y_2 \in \mathcal{S}} \left( \sum_{x_1, x_2 \in \mathcal{M}} \left( \left(\frac{1_{x_1 \in \text{proj}_{\mathcal{M}}(y_1)}}{|\text{proj}_{\mathcal{M}}(y_1)|}\right) \left(\frac{1_{x_2 \in \text{proj}_{\mathcal{M}}(y_2)}}{|\text{proj}_{\mathcal{M}}(y_2)|}\right) p_{\mathcal{D}}(x_1 \succ x_2) \times \log\left(\frac{e^{r(x_1)}}{e^{r(x_1)} + e^{r(x_2)}}\right)\right)\right) dy_2 dy_2$$

$$+ \frac{1}{|\mathcal{S}|} \frac{\lambda}{2} \int_{y \in \mathcal{S}} \left( \sum_{x \in \mathcal{M}} \left(\frac{1_{x \in \text{proj}_{\mathcal{M}}(y)}}{|\text{proj}_{\mathcal{M}}(y)|}\right) r(x)^2\right) dy$$

$$= -\frac{1}{|\mathcal{S}|^2} \int_{y_1, y_2 \in \mathcal{S}} \left( \sum_{\substack{x_1 \in \text{proj}_{\mathcal{M}}(y_1) \\ x_2 \in \text{proj}_{\mathcal{M}}(y_2)}} \left( \left(\frac{1}{|\text{proj}_{\mathcal{M}}(y_1)||\text{proj}_{\mathcal{M}}(y_2)|}\right) p_{\mathcal{D}}(x_1 \succ x_2) \times \log\left(\frac{e^{r(x_1)}}{e^{r(x_1)} + e^{r(x_2)}}\right)\right)\right) dy_2 dy_2$$

$$+ \frac{1}{|\mathcal{S}|} \frac{\lambda}{2} \int_{y \in \mathcal{S}} \left( \sum_{x \in \text{proj}_{\mathcal{M}}(y)} \left(\frac{1}{|\text{proj}_{\mathcal{M}}(y)|}\right) r(x)^2\right) dy$$

$$= f_{\mathcal{D}}(r) = -\frac{1}{|\mathcal{S}|^2} \int_{y_1, y_2 \in S} \overline{\mathcal{L}}(y_1, y_2) dy_1 dy_2 + \frac{\lambda}{2|\mathcal{S}|} \int_{y \in \mathcal{S}} \overline{r^2}(y) dy.$$

$\square$

# I. Additional Case Study Details
## I.1. Annotator Types
We use OpenAI's gpt-4o-mini model to simulate human annotators with diverse preferences. There are three populations of annotators, each of which attach a different amount of importance to seeing 'food', 'art', and 'romance' mentioned in a description of Paris. Each of the annotator types and their prevalence in the overall population are shown in the table below.

## I.2. Case Study Preference Dataset Generation
In the dataset 'Original', each preference datum is generated in the following manner. First, we randomly select (with replacement) two categories out of ['food', 'art', 'romance'] and ask OpenAI's gpt-4o-mini model to generate one response each to the prompt 'briefly describe Paris with a focus on [category]'. Next, we randomly select one annotator type with

| Alternative | Type 1 (40%) | Type 2 (30%) | Type 3 (30%) |
|---|---|---|---|
| Romance | 'really like' | 'mildly interested in' | 'very uninterested in' |
| Art | 'mildly interested in' | 'very uninterested in' | 'really like' |
| Food | 'very uninterested in' | 'really like' | 'mildly interested in' |

*Table 4.* Representation of annotator population. The prevalence of each type of annotator is indicated in the header row, while the importance that each type of annotator attaches to each category is presented in the matrix.

probability equal to their prevalence in the overall population. We then instruct the model to act as the selected annotator type and report a preference between the two responses. The dataset with a cloned alternative is generated in the same manner, except we randomly select (with replacement) two categories out of ['food', 'art', 'romance', 'restaurants'] instead of simply the first three. Note that the annotator types remain the same. Each dataset contains 1000 data points. Full prompts and sample responses for all interactions with gpt-4o-mini are given below.

**Prompt**

You are a helpful assistant. Briefly describe Paris with a focus on food, and highlight one activity.

**Response**

Paris, often referred to as the "City of Light," is renowned for its rich culinary heritage and vibrant food scene. The city is a paradise for food lovers, offering a delightful blend of traditional French cuisine and innovative gastronomy. Iconic dishes such as coq au vin, bouillabaisse, and ratatouille can be found in charming bistros and upscale restaurants alike. Paris is also famous for its patisseries, where you can indulge in exquisite pastries like macarons, éclairs, and the decadent croissant.

**Prompt**

You are a helpful assistant. Briefly describe Paris with a focus on art, and highlight one activity.

**Response**

Paris, often regarded as the capital of art and culture, is a city steeped in history and creativity. It boasts renowned museums like the Louvre, which houses masterpieces such as the Mona Lisa and the Venus de Milo, as well as the Musée d'Orsay, famous for its collection of Impressionist works. The city's streets are filled with iconic artistic sites, from the Montmartre district, once home to artists like Picasso and Van Gogh, to contemporary galleries in the Marais.

---

**Prompt**

You are visiting Paris. You really like food, are mildly interested in romance, and are very uninterested in art. You are given two answers to the question, 'Describe Paris'. Which do you prefer? Respond with only (1) or (2).

(1) Paris, often referred to as the "City of Light," is renowned for its rich culinary heritage and vibrant food scene. The city is a paradise for food lovers, offering a delightful blend of traditional French cuisine and innovative gastronomy. Iconic dishes such as coq au vin, bouillabaisse, and ratatouille can be found in charming bistros and upscale restaurants alike. Paris is also famous for its patisseries, where you can indulge in exquisite pastries like macarons, éclairs, and the decadent croissant.

(2) Paris, often regarded as the capital of art and culture, is a city steeped in history and creativity. It boasts renowned museums like the Louvre, which houses masterpieces such as the Mona Lisa and the Venus de Milo, as well as the Musée d'Orsay, famous for its collection of Impressionist works. The city's streets are filled with iconic artistic sites, from the Montmartre district, once home to artists like Picasso and Van Gogh, to contemporary galleries in the Marais.

---

**Response**

(1)

---

### I.3. Extracting Context and Neural Network Training

We approximate both the standard MLE algorithm and the weighted MLE algorithm using neural networks. Each neural network takes as input a context vector and outputs a reward value. To generate the context vectors, we first use OpenAI's text-embedding-3-small model to extract embedding vectors from the textual descriptions of Paris in each dataset. For each dataset, we then conduct principal component analysis (PCA) on the associated embedding vectors using the PCA class from sklearn.decomposition, setting the number of components equal to $1500$. The PCA-transformed data is then given as input to the neural networks. The neural network we used had 2 layers, and each hidden layer had size 32 (and output size 1). For training we used the Adam optimizer with a learning rate of $10^{-4}$ and a batch size of 512 and implemented the training using PyTorch. We trained the neural networks for 500 steps and then averaged the results over 20 runs to form the graphs included in this paper.

### I.4. Weight Estimation

To estimate the weights for the weighted MLE, we sampled 100000 random vectors within the chosen space $S$ and computed the weight for each of the alternatives in the dataset. In Figure 4 we chose $S$ to be the unit cube in the context vector space. Another natural choice of $S$ is any vector such that every coordinate of that vector is within a factor of 2 of the corresponding coordinate in one of the vectors in the observed set of alternatives. By restricting to vectors with coordinates close to those in the observed set of alternatives, this is potentially a more suitable choice for $S$ than choosing $S$ to be the entire unit cube. This is because the unit cube may include many alternatives that are not reasonable answers to the question. While there are arguments for many different choices of $S$, we also analyze this choice of $S$ to demonstrate the robustness of the weighted MLE. The results for this choice of $S$ are shown below in Figure 5, and the weighted MLE remains robust to the addition of approximate clones.

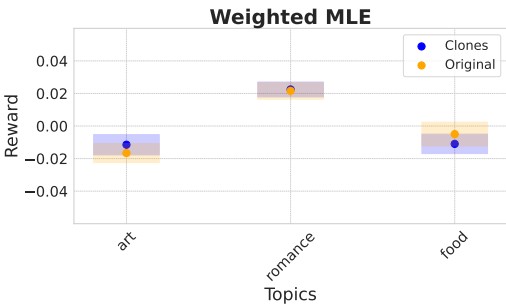

*Figure 5.* Results for the weighted MLE when $\mathcal{S}$ is chosen as any vector such that every coordinate is within a factor of $2$ of one of the observed coordinates. The yellow points show the average value of the weighted MLE reward function for different topics when trained on dataset 'Original'. The blue points show the same but when trained on dataset 'Clones'. In both cases, the reward function has the highest value for romance, demonstrating the robustness of the weighted MLE.

