# OpenReview forum: "Clone-Robust AI Alignment"
_ICML.cc/2025/Conference — ICML 2025 poster_

### Official Review · Reviewer_b1RX · 2025-03-04

**Overall Recommendation:** 3

**Summary:**

The paper evaluate the robustness of current RLHF algorithms in the presence of approximate clones and develop RLHF algorithms to enhance the robustness regarding this.

**Claims And Evidence:**

Yes, I think most of the claims made in the submission clear and convincing. However, the empirical experiments (case study) are not sufficient enough for me.

**Essential References Not Discussed:**

N/A

**Experimental Designs Or Analyses:**

The paper is mostly about theory. See Claims And Evidence part.

**Methods And Evaluation Criteria:**

See Claims And Evidence part.

**Other Comments Or Suggestions:**

N/A

**Other Strengths And Weaknesses:**

See Questions For Authors

**Questions For Authors:**

1.  why the focus of the paper is RLHF rather than reward model itself?
2.  I am not quiet familiar with the area. Could you please explain briefly how the theoretical framework can guide the practical implementation of training real LLMs?

**Relation To Broader Scientific Literature:**

The paper provides a good vision for the current understanding of RM/RLHF.

**Theoretical Claims:**

I didn't check all the proofs in detail but the theorems provides seem to be reasonable and sounding.

---

> ### Author Rebuttal · Authors · 2025-03-31
>
> Thank you for your comments! Below we address your specific questions:
>
> > why the focus of the paper is RLHF rather than reward model itself?
>
> Our paper focuses on the step of the RLHF pipeline that takes as input a  preference dataset and outputs a reward model. In RLHF, this reward model is then used to fine-tune LLMs. However, the focus of our paper (robustness to approximate clones) is a desirable property of the process of going from preference data to a reward model. We highlight implications for RLHF because RLHF is currently a very common application of reward modeling from preference data, but our results could be used in any application of reward modeling.
>
> > I am not quiet familiar with the area. Could you please explain briefly how the theoretical framework can guide the practical implementation of training real LLMs?
>
> To use our framework to train LLMs in practice, one would execute the following two steps. First, learn a reward function by computing the weighted MLE on the given pairwise comparisons; this is the part addressed (and potentially improved) by our paper. The second step is a standard policy optimization step, where (roughly speaking) a given base LLM is fine-tuned to maximize the expected reward from step 1, for example, through proximal policy optimization (PPO). Since the second step simply takes as input a base LLM and a reward model, the output of our weighted MLE can be directly plugged into this heavily studied pipeline.

---

### Official Review · Reviewer_CwdH · 2025-03-13

**Overall Recommendation:** 4

**Summary:**

The paper considers axiomatic AI alignment. More precisely, the paper is about Reinforcement Learning with human feedback (RLHF). As motivated by Conitzer et al. (2024), consistency with respect to clones is an interesting property for RLHF algorithms. In this paper, each alternative is identified with its context, i.e., some $d$-dimensional real vector that lies in some infinite set of finite volume $S$. Roughly speaking, the goal is to aggregate the utility functions of individual voters (annotators) into a collective utility function. The catch is that we are only given a finite set $M\subseteq S$, query samples of the voter preferences over $M$, and want this aggregated utility function to be clone-proof.
Each annotator has a utility function $r$ over the alternatives in $M$, and whenever asked to compare two alternatives, says that they prefer alternative $a$ over $b$ with probability $ e ^ ( r(a) ) /  [ e ^ ( r(a) ) + e ^ ( r(b) ) ] $, known as the Bradley-Terry (BTL) model.
Queries are (two)-alternative subsets of the form \{a,b\}, and we denote by $Q$ a (multi)-set of queries where each possible query is contained at least once. For each query $q \in Q$, we choose an annotator uniformly at random and obtain a sample of the annotator’s preference for that query. In total, we obtain a random dataset $D$ consisting of all queries in $Q$ and the respective responses. Given the query set $Q$ and the resulting dataset $D$, the goal is to find a social utility function $r$ that best models the collective preference of the annotators.

The first result [Theorem 2.3] states that no algorithm can always output a collective preference $r$ that is equal to the mean reward function, i.e., E_i[r_i]. This impossibility already holds true for two voters. Next [Theorem 2.5], the authors establish a relation between the average win rate and the regularized MLE, which is defined as the utility function $r^D = r$ minimizing

0.5 \lambda  \sum_{x\in M} r(x)^2  - \sum_{x_1,x_2\in M} p_D(x1>x_2) log( e^(r(x_1) / [ e^(r (x_1) + e^( r (x_2) ) ] ).

The authors then introduce their core axiom, robustness to approximate clones, which says that for each $\delta>0$ there should be an epsilon>0 such that adding an alternative that is of distance at most epsilon to an existing alternative changes the reward functions by all alternatives by at most delta, and the utility for the almost-clone is also similar to the original utility of the alternative that it almost clones. Clearly, by being Borda-like, MLE violates robustness to approximate clones [Theorem 3.2], as it even fails robustness to precise clones.

To counteract this phenomenon, the authors introduce a Voronoi-approach to define a distribution $w_D$ over $D$: each alternative in $S$ is projected to its closest candidate(s) in $D$ and then, for the weight of $y\in \mathcal M$, $w_D(y)$ is calculated using the volume of the set of all $x\in S$ that are projected onto $y$ [Definition 4.1].

The main result of the authors is that the reweighted MLE satisfies robustness to approximate clones [Theorem 4.2].
Analogously to Thm 2.5,  the authors present an identity involving the weighted average win rate and the weighted  MLE estimator [Theorem 4.4], implying that the ordering induced by the weighted MLE estimator is equal to the weighted average win rate of the alternatives [Corollary 4.5]. Then, the authors argue that the weighted MLE approach is an approximation of the MLE over S [Theorem 4.6] (This essentially boils down to swapping some sums and integrals).
The authors then discuss a synthetic case study that illustrates the susceptibility of MLE to clones, while the weighted MLE unsurprisingly performs better.

**Claims And Evidence:**

Claims are of a mathematical nature and supported by proofs

**Essential References Not Discussed:**

N/A

**Experimental Designs Or Analyses:**

I did not.

**Methods And Evaluation Criteria:**

The proposed weighted MLE is natural and makes sense for the problem at hand.

**Other Comments Or Suggestions:**

The authors consider weighting alternatives in \mathbb R^d within a set $S$ of bounded volume. When moving from $S$ to the unbounded $\mathbb R^d$, there is a paper by Berriaud and Wattenhofer that considers this version of the problem. Most notably, I see some similarities between their axiom 6 (alpha-locality under the addition of clones) and the here-considered Definition 3.1 (robustness to approximate clones). Further, both papers use the idea of utilizing projections for integrals and their formula for the $g$ function in Section $4$ seems to resemble the approach taken in this paper for the weighted MLE.
As the paper came out after the ICML deadline this is of course concurrent work and not relevant to the judgment of the paper, however, in case of acceptance, the paper should still compare to it (to keep the academic record complete).

## update after rebuttal

Thank you for the nice rebuttal!

**Other Strengths And Weaknesses:**

The findings of the paper would be a good addition to the conference. I like the question asked by the authors and think it is both quite natural and interesting. Further, I am a big fan, of recent works combining social choice theory and AI alignment and think that this paper provides a very interesting twist on it. For the most part, the paper is also well written, and nice to read, even for a reader who is not necessarily well versed in the literature on LLMs.

One criticism is that currently, the precise properties of $S$ are not defined in the preliminaries, there is no mention of $S$ being Borel-measurable or of finite(!) volume, which confused me for quite a bit and seems to be crucial for the paper if I am not mistaken. In several parts, the paper could also be more non-expert friendly, e.g., KL divergence is mentioned but not defined on page 4.

**Questions For Authors:**

1. It is unclear to me why it is desirable that the average win rate is the sum of the empirical win rate and the reward function itself. Is there some motivation for this?
2. In social choice, there is a strengthening of independence of clones called composition consistency. Roughly speaking, if one replaces an alternative with a component, then the probability of the alternative gets distributed to the alternatives in the component is precisely proportional to how the probabilities would have been if the component was viewed in isolation. Have you thought about whether this notion could make sense in this context?

**Relation To Broader Scientific Literature:**

The study of consistency w.r.t. to clones is indeed an important task for AI alignment, see e.g., Conitzer et al.
The proposed approach using Voronoi diagrams seems sensible and suits this task

**Theoretical Claims:**

The proofs of Theorem 2.3 and Theorem 4.6 are sound.

---

> ### Author Rebuttal · Authors · 2025-03-31
>
> Thank you for your helpful comments and feedback! Below we address your specific questions:
>
> > It is unclear to me why it is desirable that the average win rate is the sum of the empirical win rate and the reward function itself. Is there some motivation for this?
>
> The original motivation for this result (Theorem 2.5) was to give better intuition for why the ranking induced by the MLE estimator is the same as the Borda Count ranking (which follows directly from the relationship between the estimated win rate and the empirical win rate). A secondary benefit of Theorem 2.5 is that it gives an additional motivation for the MLE solution. A natural alternative to using MLE estimation is to find the reward function that best matches the empirical win rates; Theorem 2.5 says that the BTL MLE solution is (almost) equivalent to matching empirical win rates.
>
>
>
> >In social choice, there is a strengthening of independence of clones called composition consistency. Roughly speaking, if one replaces an alternative with a component, then the probability of the alternative gets distributed to the alternatives in the component is precisely proportional to how the probabilities would have been if the component was viewed in isolation. Have you thought about whether this notion could make sense in this context?
>
>
> Mapping composition consistency to RLHF is difficult for a few reasons. In social choice, composition consistency is defined by running a social choice function twice on a set of rankings: once with the clone sets grouped together and once on the winning clone set. This is possible because social choice functions map rankings (over alternatives) to alternatives, so they can be applied iteratively. In RLHF, however, we are mapping pairwise comparisons to a reward function, so we would have a type error if we tried to use an analogous definition.
>
> Composition consistency also may not be a desirable property for RLHF. In contrast to traditional social choice, RLHF assigns a reward to every alternative, and therefore the order of alternatives does not matter as much as the actual rewards. Specifically, for a set of approximate clones, we do not care about the order of the clones (like in composition consistency), but instead we want that the clones all have similar reward values. In fact, for any output reward function that is continuous, any set of approximate clones will also have approximately the same reward as desired. Therefore, composition consistency does not seem desirable/necessary in RLHF as long as the output reward function is continuous.
>
> > One criticism is that currently, the precise properties of $S$ are not defined in the preliminaries, there is no mention of being Borel-measurable or of finite(!) volume, which confused me for quite a bit and seems to be crucial for the paper if I am not mistaken. In several parts, the paper could also be more non-expert friendly, e.g., KL divergence is mentioned but not defined on page 4.
>
>
> Thank you for pointing this out. We certainly agree that $S$ must be finite and measurable, and we will add more specific properties of $S$ in the model section for the final version of the paper. We will also make sure to define any more niche technical terms used throughout.
>
> > ... there is a paper by Berriaud and Wattenhofer that considers this version of the problem...
>
> Thank you for bringing this to our attention! We will be sure to discuss this work in the final version of our paper.

---

> > ### Comment · Reviewer_CwdH · 2025-04-02
> >
> > Thank you for the nice response!

---

### Official Review · Reviewer_R2Xe · 2025-03-13

**Overall Recommendation:** 3

**Summary:**

The paper addresses a key challenge in LLM alignment, that of making sure that the RLHF model is unbiased. Specifically, authors show that the distribution of data used to train the model can have a significant impact on how the RLHF model behaves, and as such it is prone to intentional or unintentional biases. This usually happens when the dataset is biased (authors talk about duplicate or near duplicate pieces of data).

The authors propose the concept of "clone-robustness" meaning the the model training is immune to data clones present in the dataset. This is done by using a weighted MLE algorithm that assigns lower weights to alternative data points that are similar to other data points.

**Claims And Evidence:**

Yes, most claims are well supported, for example

"standard RLHF is not robust to clones"
1. Theoretical proof is provided via theorem 3.2
1. The case study in section 5 also supports the claim above

"weighted MLE ensures robustness to approximate clones"
1. Theoretical proof is provided via theorem 4.2
1. Also backed up by the case study in section 5

Couple of unproven claims:

Generalizability of Weighted MLE
1. Tested only in a narrow set of scenarios (describe Paris), which makes the real world effectiveness unclear
2. A specific weighing scheme is discussed, but its unclear whether alternative schemes might perform better or worse

**Essential References Not Discussed:**

Christiano et al. (2017) : "Deep reinforcement learning from human preferences" (NeurIPS 2017): This foundational paper introduced RLHF, demonstrating that LLMs can learn from pairwise human preferences. The authors critique RLHF's vulnerability to dataset biases, but do not cite Christiano et al. (2017), where these concerns first emerged.

**Experimental Designs Or Analyses:**

### Strengths

1. Directly Tests Clone Robustness: The controlled introduction of cloned responses effectively isolates the impact of near-duplicates on RLHF training.
1. Uses Embedding-Based Similarity Measures: The use of OpenAI’s text-embedding-3-small model to represent response similarities is a reasonable approximation of how RLHF embeddings work in real-world AI training.
1. Quantitative Analysis with Win Rate Comparisons: The study evaluates how reward scores shift across different topic categories (food, art, romance), with error bars for variance.

### Weaknesses

1. Use of LLMs as Annotators Instead of Human Feedback: The study simulates human preference data using an LLM (GPT-4o-mini) instead of actual human annotators.
1. Narrow Scope of Dataset (Single Prompt: “Describe Paris”): The experiment only tests one question, meaning results may not generalize across different types of RLHF tasks (e.g., safety-critical alignment, long-form reasoning)
1. No Evaluation on Real-World RLHF Datasets: The datasets used are synthetic, and the paper does not benchmark performance on real RLHF datasets.
1. Limited Statistical Analysis: The paper visually presents reward differences (e.g., Figure 3 & 4) but does not conduct rigorous statistical significance tests.
1. No Robustness Testing for Weighted MLE with Alternative Weighting Schemes: The experiment only evaluates one version of Weighted MLE with a fixed weighting function.

**Methods And Evaluation Criteria:**

The methods used (theoretical proof + experimental validation) sound correct for the problem at hand. However, the scope of the study is quite narrow, which means that it is unclear how the weighted MLE would perform on a diverse set of scenarios.

The study simulates human preferences using an LLM as an annotator. An LLM might not accurately capture human's method of annotations. Similarly, the diversity of annotator preferences is modeled using fixed categories.

One key missing piece is the lack of validation of the weighted methods against some standard RLHF datasets from industry or academia.

**Other Comments Or Suggestions:**

None

**Other Strengths And Weaknesses:**

The key contributions / strengths are
1. formal definition of robustness to approximate clones
1. proving standard MLE is not clone robust
1. Proposing and proving that weighted MLE works

Key gaps are
1. lack of validation / benchmarking against real world RLHF
2. lack of discussion on weighing schemes
3. lack of comparison with alternative solutions for clone robustness

**Questions For Authors:**

1. Have you tested Weighted MLE on real-world RLHF datasets? Please share results if so.
1. How does Weighted MLE perform across different types of RLHF tasks (e.g., factual questions, multi-turn dialogue)?
1. Could you provide real-world examples where approximate clones have distorted RLHF in deployed AI systems?

**Relation To Broader Scientific Literature:**

Tideman (1987):  This paper builds upon Tideman's introduction of "independence of clones" by adapting it from voting theory to RLHF, proposing a new algorithm (weighted MLE) that ensures robustness to approximate clones.

Elkind et al. (2010, 2012): The authors extend Elkind et al.'s studies on manipulation through cloning by applying similar ideas to RLHF, highlighting vulnerabilities in standard RLHF algorithms and motivating their new robust solution.

Conitzer et al. (2024): This paper elaborates on Conitzer et al.'s suggestion that independence of clones is important for RLHF, providing concrete examples and a new algorithm addressing this issue.

Xu et al. (2023): The authors share Xu et al.'s concern about duplicates in RLHF datasets but extend their results beyond dichotomy models and three-way comparisons to standard pairwise comparisons.

Siththaranjan et al. (2023): This paper extends Siththaranjan et al.'s insights about regularized MLE and average win rates by proving a stronger theoretical relationship, and builds upon their impossibility result for diverse preferences by showing an even stronger impossibility result.

**Theoretical Claims:**

No, I did not look at the theoretical proofs in detail.

---

> ### Author Rebuttal · Authors · 2025-03-31
>
> Thank you for your comments! Below we address your specific questions:
>
> > Have you tested Weighted MLE on real-world RLHF datasets? Please share results if so. How does Weighted MLE perform across different types of RLHF tasks (e.g., factual questions, multi-turn dialogue)? Could you provide real-world examples where approximate clones have distorted RLHF in deployed AI systems?
>
> This paper is primarily a theoretical contribution to the field of RLHF, as we propose a theoretical property and prove results about that property for current and new algorithms. While we included the case study as a proof-of-concept for the proposed weighted MLE, more intensive experiments using the weighted MLE in different applications is beyond the scope of our paper. We do however think this is a very important topic for future work, and we hope to provide sufficient motivation and information for practical applications of the weighted MLE in the future.
>
>
> > Key gaps are
> > - lack of discussion on weighing schemes
> > - lack of comparison with alternative solutions for clone robustness
>
> Because robustness to clones in the context of RLHF was introduced in this paper, there is no previous work that has alternative solutions for this problem. There are voting rules from traditional social choice that are independent of clones, and these could potentially be adapted to the RLHF setting. However, such solutions would be very different than the current MLE estimation and may be less practical. We briefly mention other weighting schemes at the end of the discussion, specifically that the $w(\cdot)$ function can potentially be replaced with other functions that upweight more unique alternatives. We are happy to include more discussion on both of these points in the final paper.

---

### Official Review · Reviewer_MLTh · 2025-03-13

**Overall Recommendation:** 4

**Summary:**

This paper mainly focus on the problem of unbalanced input datasets in RLHF, which is caused by adversarial manipulation or inadvertent repetition. The key motivation is to make RLHF robust towards non uniformly distributed datasets. Inspired by social choice theory, they introduced robustness to approximate clones, a desirable property of RLHF algorithms which requires that adding near-duplicate alternatives does not significantly change the learned reward function.

**Claims And Evidence:**

They show that the standard RLHF algorithm based on regularized maximum likelihood estimation (MLE) fails to satisfy this property. In contrast, a weighted MLE can alleviate this problem.

**Essential References Not Discussed:**

NA

**Experimental Designs Or Analyses:**

The experiment is a bit toy setting.

**Methods And Evaluation Criteria:**

The voting rule robust towards adding duplicates of alternatives is important. The authors alims this as "satisfy independence of clones". Informally, a voting rule is independent of clones if after adding an alternative a', which is equivalent to another alternative a, the output of the voting rule does not change. In RLHF, there do exist "approximate clones", namely two alternatives which are very close by a given distance metric and for which all annotators have very similar values, where the distance metric depends on the nature of the alternatives.

The proposed new training objective is simple, via, down-weighting alternatives that are similar to other alternatives (and therefore provide less new information) and up-weighting alternatives that are different than other alternatives (and therefore provide more new information)

**Other Comments Or Suggestions:**

NA

**Other Strengths And Weaknesses:**

The presentation is great.
The motivation towards studying unbalanced dataset is clearly explained with examples provided.
There is one question towards the necessity of removing near duplicates.

**Questions For Authors:**

"Fundamentally, the mandate of RLHF algorithms is to solve a preference aggregation problem". Could you please share your insight on why it is related to "aggregation"? In my opinion, it is just to generalize preference even given conflict preferences existed.

One question towards "near duplicate": Sometimes these near duplicates are necessary in practice, because some topic do have a heavy weight among the entire topic distribution.

**Relation To Broader Scientific Literature:**

This work can give some inspiration to the practical reward model training pipeline optimization, especially handling the data unbalance.

**Theoretical Claims:**

As an extension, if n>2 in Theorem 2.3, will the proof still stand?

---

> ### Author Rebuttal · Authors · 2025-03-31
>
> Thank you for your helpful comments and feedback! Below we address your specific questions:
>
> > As an extension, if n>2 in Theorem 2.3, will the proof still stand?
>
> Yes, the results do extend for $n > 2$. Learning a reward function can only become information theoretically harder for $n > 2$ because there are more variables that need to be estimated. Therefore, the same impossibility result holds for larger $n$ as well.
>
> > "Fundamentally, the mandate of RLHF algorithms is to solve a preference aggregation problem". Could you please share your insight on why it is related to "aggregation"? In my opinion, it is just to generalize preference even given conflict preferences existed.
>
> In the field of social choice, preference aggregation refers to the problem of taking voter preferences that are potentially conflicting, and *aggregating* them into a single output that captures the voters' preferences. Mapping this onto RLHF, the annotators may have conflicting preferences, and the goal is to find a single reward function that captures the annotators' preferences. Therefore, "aggregation" in RLHF refers to the aggregation of all of the individual annotator preferences into one single reward function that can be used for LLM tuning.
>
>
>
> > One question towards "near duplicate": Sometimes these near duplicates are necessary in practice, because some topic do have a heavy weight among the entire topic distribution.
>
> This is a great point, and we thought about this a lot while writing the paper. In practice, we definitely expect that some topics would be more heavily weighted in the overall topic distribution. In fact, having near duplicates in the dataset can be good, because more comparisons involving two common response topics gives a better estimate of the annotators' preferences between these two topics. However, we believe it is undesirable for an RLHF algorithm to reward or punish a topic based on its weight in the topic distribution. Instead, we want the final reward value for an answer topic to only depend on the annotators' preferences for that topic (and not on the topic distribution in the data set). In summary, we don't think that near duplicates are inherently bad -- we just want the final reward function to be stable regardless of the topic distribution for the observed preference data set.

---

### Decision · Program_Chairs · 2025-05-01

**Decision:**

Accept (poster)

**Comment:**

This paper addresses a key limitation in RLHF systems—their vulnerability to "clone attacks," where repeated or near-duplicate preferences can bias the learned policy. It introduces a theoretically grounded and practically implementable solution using a weighted MLE objective that remains interpretable while being robust to such clones. The paper combines tools from RLHF, social choice theory (notably clone independence), and theoretical learning guarantees. Reviewers agree that the motivation is important, the formulation is elegant, and the results are compelling both in theory and experiment.

The paper addresses a novel and important failure mode in RLHF—vulnerability to clone preferences—through a principled, interpretable weighted MLE framework. Theoretical guarantees and an illustrative case study support its effectiveness. The formulation is clear, well-motivated, and applicable to real-world alignment problems. Despite some limitations in scope and experimental scale, the work makes a clear and meaningful contribution to the field of RLHF and alignment theory, and meets the standards for inclusion at ICML. Please incorporate all the reviewer's feedback in the final version.